# BONGARDS AT THE BOUNDARY OF PERCEPTION AND REASONING: PROGRAMS OR LANGUAGE?

## ABSTRACT

Vision-Language Models (VLMs) have made great strides in everyday visual tasks, such as captioning a natural image, or answering commonsense questions about such images. But humans possess the puzzling ability to deploy their visual reasoning abilities in radically new situations – a skill rigorously tested by the classic set of visual reasoning challenges known as the Bongard problems. We present a neurosymbolic approach to solving these problems: given a hypothesized solution rule for a Bongard problem, we leverage LLMs to generate parameterized programmatic representations for the rule and perform parameter fitting using Bayesian optimization. We evaluate our method on classifying Bongard problem images given the ground truth rule, as well as on solving the problems from scratch.

## 1 INTRODUCTION

Visual reasoning lives at the boundary of perception and thinking. For this reason, it is a central object of study in artificial intelligence and cognitive science (Ullman, 1987; Andreas et al., 2016, *inter alia.*). Vision-Language Models (VLMs) have made great strides in everyday visual tasks, such as captioning a natural image, or answering commonsense questions about such images. But humans possess the puzzling ability to deploy their visual reasoning abilities in radically new situations, far outside everyday experience (LeGris et al., 2024), ranging from visualizing curved geometry to making sense of cubist artwork. How close is AI to capturing these abilities?

As a microcosm of this broadly-general visual reasoning, we turn to the Bongard problems (BPs), a few-shot visual classification dataset from 1970 (Bongard & Hawkins, 1970). Despite their age, the BPs have been studied by each successive wave of AI, from symbolic to probabilistic to neural to LLMs (Wüst et al., 2025; Depeweg et al., 2024; Foundalis, 2006; Saito & Nakano, 1996; Nie et al., 2020; Maksimov, 1975). The BPs involve perceiving novel visual features on-the-fly, such as the 'neck' in Figure 1 A(i), and then reasoning about those features to discriminate between two categories. Unlike other visual reasoning tasks such as Raven's Progressive Matrices, the essence of a BP is to fluidly define new perceptual primitives for each problem, which forces learners to rapidly generalize precisely at the boundary of perception and reasoning.

Through a close examination of the BPs, our study explores how two different modes of reasoning – reasoning over programs and over natural language – can complement each other and push us closer to a solution for difficult concept learning tasks. Solutions to the BPs combine high-level conceptual reasoning that should benefit from VLMs' vast pretraining data with geometric reasoning that requires a level of precision characteristic of programs, motivating us to pursue a hybrid approach that could leverage the strengths of both program synthesis and reasoning over natural language.

In total, we contribute the following:

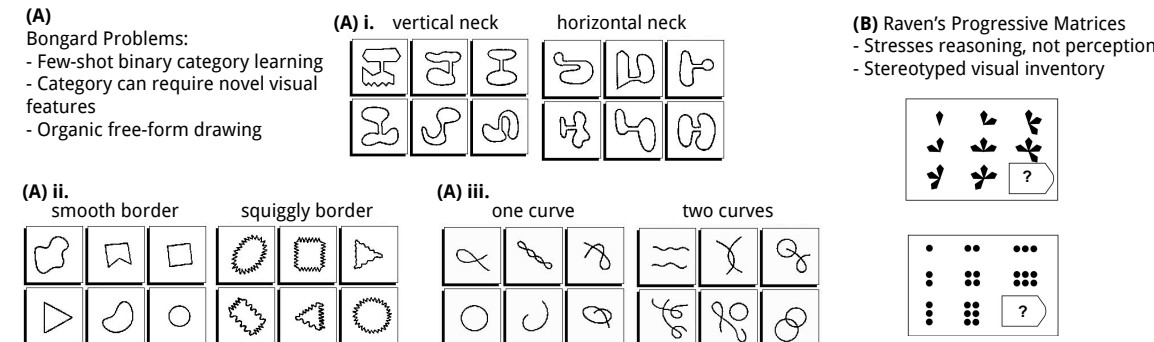

Figure 1: (A) Example Bongard problems, each of which consists of 6 positive examples (left drawings) and 6 negative examples. The natural language descriptions shown above each category are not provided to the learner. (B) In comparison, the well-known Raven's Progressive Matrices trade perceptual richness for deeper logical composition.

1. An evaluation of State-of-the-Art vision language models on two distinct tasks: classifying BP images when the ground truth rule is present and solving the BPs by eliciting ground truth rules.

2. A new method that leverages the synthesis of parameterized programs alongside natural language reasoning to enable the selection of correct natural language rules.

3. An analysis of human data to quantify and compare vanilla VLMs and our approach against human performance.

## 2 THE BONGARD PROBLEMS

The Bongard problems (BPs) are a set of abstract visual reasoning puzzles first presented in Mikhail Bongard's 1970 book *Pattern Recognition* (Bongard & Hawkins, 1970). Each BP consists of twelve total images – six examples of a positive concept and six examples of a negative concept. The negative concept may be the simple negation of the positive concept, or it may be a different concept entirely. Furthermore, unlike reasoning puzzles like Raven's Progressive Matrices (Raven & Raven, 2003), there is no sequential or causal relationship between the different positive and negative examples; each image is independently an example of the underlying positive or negative concept. A solution to a BP is a natural language rule that completely separates the positive examples from the negative examples (Hofstadter, 1999); for instance, "has a smooth border" would be an acceptable solution to BP #9 shown in Figure 1. Although BPs are intentionally designed puzzles created with particular solutions in mind, multiple correct solutions may exist, as long as they properly separate the positive and negative examples (Depeweg et al., 2024).

The concepts tested by the BPs are generally abstract and geometric in nature. Some of these concepts, such as "triangles" vs. "quadrilaterals" in the solution of several BPs, may already be familiar to the solver. Others, like the aforementioned solution to BP#19 (horizontal "neck" vs. vertical "neck"), involve a combination of concepts that the solver is not likely to have encountered before. Furthermore, the particular features of an image that are relevant to the correct solution vary greatly between BPs, making them resistant to solution via image preprocessing.

The original collection of Bongard problems were designed to be solved by humans in sequence, and give a natural curriculum that gradually increases problem complexity while introducing new concepts one-by-one. The system that we describe exploits the curriculum structure of the Bongard problems.

## 3 RELATED WORK

**Human Performance on BPs.** The BPs test a wide variety of unfamiliar concepts, but human performance is generally strong in spite of these challenges: Wüst et al. (2025) find that the average human solves approximately 47 Bongards for BPs (#2-#100), while the top 5 human solvers averaged approximately 63 problems.

**Automatically Solving BPs.** There is a long history of attempting to solve the BPs in AI (Foundalis, 2006; Maksimov, 1975; Saito & Nakano, 1996), but the problems have become relevant once more due to state-of-the-art AI systems' struggle with human-solvable reasoning and perception tasks. Recent attempts to solve the BPs have included Bayesian inference over a formal language (Depeweg et al., 2024) and program synthesis coupled with inductive logic programming (Sonwane et al., 2021), as well as attempts to reimagine the BPs as a more traditional ML-based classification task (Raghuraman et al., 2023), an RL task setting (Youssef et al., 2022), or as inspiration for new datasets that test the reasoning capabilities of VLMs (Nie et al., 2020; Wu et al., 2023). An evaluation of VLMs on the Bongard problems was notably absent until Wüst et al. (2025), which tested a number of VLMs on solving the Bongards, as well as a number of variations on the Bongard task. The best tested model, o1 (Jaech et al., 2024) solved 43 BPs – a significant improvement over previous attempts to solve BPs automatically that still falls short of average human performance.

**Reasoning with Code.** As LLMs emerge as ever more capable tools for code generation (Li et al., 2022; Novikov et al., 2025) and as they continue to struggle with complex reasoning tasks, many different systems for generating executable code for solving reasoning problems have been proposed (Gao et al., 2023; Li et al., 2023). For these systems, the answer to a reasoning problem is an executable program that, when run on an appropriate input, produces the desired answer. When solving mathematical or algorithmic reasoning problems, reasoning using code has the clear advantage of being exact, interpretable, and (in most cases) deterministic. However, formal programs have an inherent expressivity problem: there are reasoning problems that can be easily formulated in natural language yet are difficult or impossible to express in code.

**Induction and Transduction.** The strengths and weaknesses of reasoning over both formal programs and natural language indicate room for hybrid approaches.Li et al. (2024) formulate the problem as a question of induction vs. transduction. Induction, here corresponding to the program induction approach to reasoning, is defined as the paradigm in which, before predicting outputs for the test examples, the learner must explicitly construct a function that produces the correct outputs given the training examples as inputs. On the other hand, transduction, here corresponding roughly to test-time training or the Chain of Thought (CoT) (Wei et al., 2022) approach to reasoning, outputs a prediction for the test examples given the training examples, without explicitly searching for a latent function. For the Abstraction and Reasoning Corpus (ARC) (Chollet, 2019), another task which, like the BPs, requires pattern recognition and conceptual generalization, the two reasoning paradigms were found to be complementary (Li et al., 2024), with induction and transduction excelling at different problems. Drawing inspiration from this work, we also combine programmatic and CoT reasoning to solve the BPs, in the hope that the two methods will complement each other.

## 4 METHOD

We experiment with two different tasks:

1. **Verification**: Given the other 5 positive/negative image pairs for a BP as training examples *along with the ground truth rule*, correctly classify the held out positive/negative image pair.

2. **Solution**: Given all BP images, output a rule that correctly distinguishes positive from negative ones.

In order to more accurately measure performance on the verification task, we repeat the task six times, holding out the positive and negative images at index $0$ to $5$ exactly once.

## 4.1 SYSTEM OVERVIEW

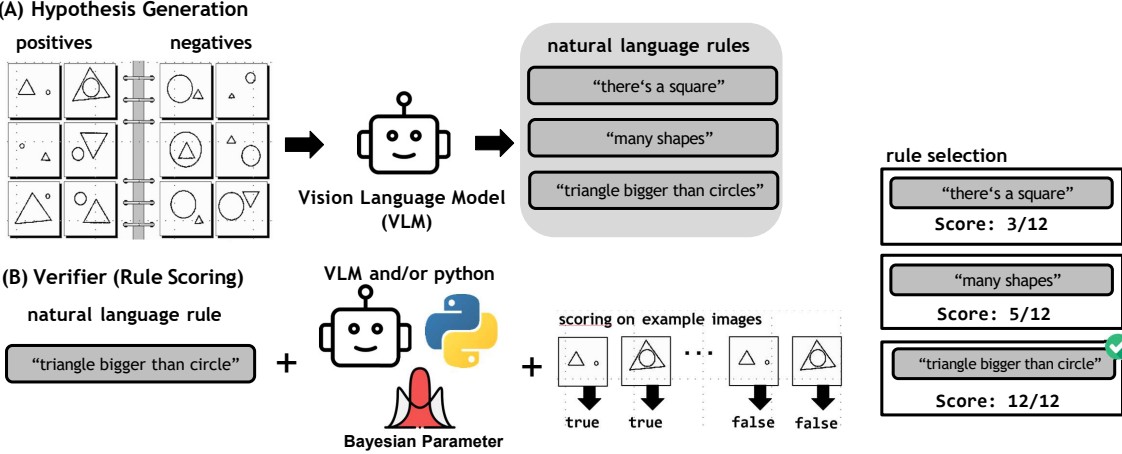

Figure 2: Our system comprises (A) a hypothesis generator that samples possible rules from a VLM and (B) a verifier that combines natural language and code to score and select the best rules.

We propose a BP solver with two main components: a hypothesis generator that generates several possible rules and a verifier that can determine which of these hypothesized solutions are correct (Figure 2). The verification task tests the verifier only: the ground truth rule and BP images are provided as input to the verifier, which then scores that rule. In contrast, for the solution task, our hypothesis generator produces several candidate rules, which are then scored by the verifier.

### 4.1.1 HYPOTHESIS GENERATOR

We sample possible solutions from a VLM, providing as input all positive and negative images for the BP and three sample rules drawn from other BPs. Because the Bongard problems were designed to be solved by humans sequentially, and have a natural curriculum ordering, we sample 6 rules given the rules of the three previous BPs as in-context examples. For increased diversity we sample another 6 rules with the solutions of three random BPs as in-context examples.

### 4.1.2 VERIFIER

The correctness of these hypothesized rules is judged by the verifier, which combines natural language and Python programs to score each proposed rule. Given a set of training examples $X_{train}$, for each image-label pair $(x_{train}, y_{train})$ and a rule, the verifier first attempts to synthesize a program $\pi$ such that $\pi(x_{train}) = y_{train}$. A candidate program's score is then

$$score(\pi) = \sum_{(x_{\text{train}}, y_{\text{train}}) \in X_{\text{train}}} \frac{\mathbb{1}\{\pi(x_{train}) = y_{train}\}}{|X_{train}|}$$

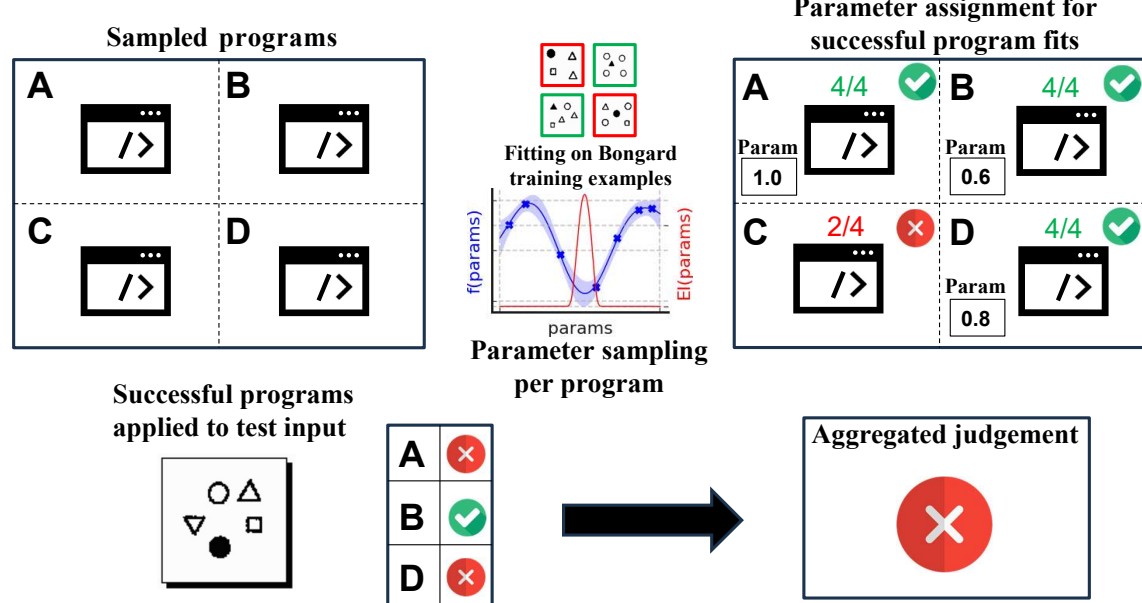

Figure 3: Overview of the program synthesis module of our verifier. Programs are sampled and undergo parameter fitting via Bayesian optimization. If we can successfully optimize programs (i.e., find programs that score at least 0.9 on the training examples), the highest-scoring programs are evaluated on the test examples, with the majority label from these evaluations serving as the output label.

Let $P$ denote the set of all candidate programs. The set of *accepted programs* $A$ is defined as

$$A = \{\pi \in P \mid score(\pi) = \max_{\phi \in P}(score(\phi)) \land score(\pi) \geq 0.9\}$$

If $A$ is nonempty, then programs will be used to determine the labels of the test images, as described in the following sections. Otherwise, the verifier switches to the CoT approach, using as the output label the result of prompting the VLM with positive and negative examples provided as in-context examples.

### 4.1.3 VERIFICATION WITH PROGRAMS

The verifier takes as input 5 positive-negative pairs of training examples, suggested *method stubs* (Appendix B), and in-context examples . For the in-context example, we use Retrieval Augmented Generation (RAG) (Lewis et al., 2020) in order to improve the quality of VLM-generated code. We generate a set of programs for 59 different Bongard problems (Appendix C) and use embedding similarity between the currently proposed rule[1] and the ground truth rules associated with each of the 59 problems to select the most relevant example and provide it as the in-context example in our prompt. $n$ programs are sampled with the VLM provided with these inputs.

For the inductive component of our verifier, the objective is to synthesize a $classify\_image$ method that, given an input image (along with additional parameters that will be explained in the following section), will return the image classification ("POSITIVE" or "NEGATIVE").

---

[1]For the verification task, this will always be the the ground truth rule for the current problem.

### 4.1.4 PARAMETERIZED PROGRAMS AND OPTIMIZATION

Humans have an intuitive idea of when an image belongs to a particular concept, but expressing this as a rule in a general-purpose formal language is challenging. For example, in BP #11 spotting the difference between 'elongated' and 'compact' shapes is not overly challenging, nor is even calculating the 'circularity' of an object or observing that the class of 'compact' objects is more similar to circles (see Figure 4). However, the exact numerical dividing line between 'elongated' and 'compact' figures would be exceptionally difficult to determine through guesswork, making it impractical to require our VLM to attempt to synthesize this value along with the rest of the program. Instead, we leverage the pretraining knowledge of the VLM to specify a range of possible values for a given parameter and perform just 15 steps of Bayesian optimization (Frazier, 2018; Shahriari et al., 2015) to find the value that maximizes the program score. The parameters that require optimization are determined by the VLM and are given as additional inputs to the synthesized 'classify_image' function. If optimizing a given program does not result in a perfect score on the training examples, the VLM is prompted to revise the programs once. These new programs undergo the optimization process once more.

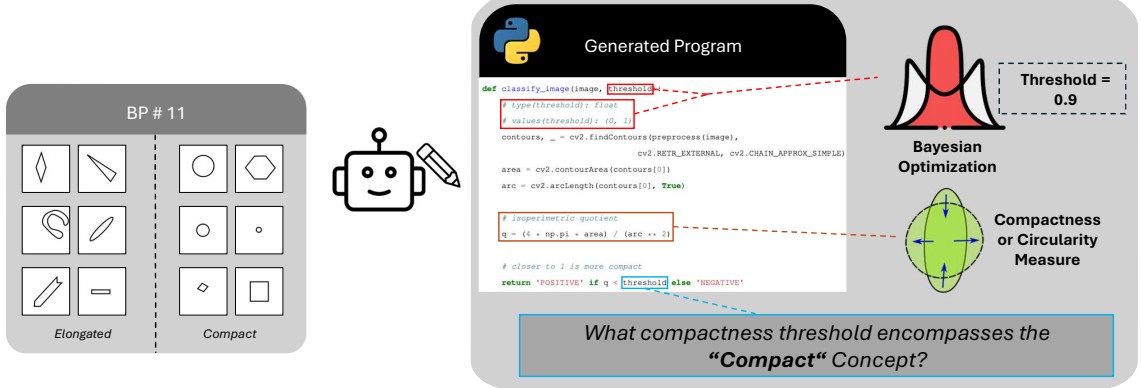

Figure 4: To solve BP#11 ('Elongated' vs. 'Compact'), the critical but difficult-to-synthesize parameter $threshold$ is optimized via Bayesian optimization.

Should the verifier successfully synthesize one or more programs that obtain a score $\geq 0.9$ on the training data, we obtain an output classification for the each held-out example by running all programs that have achieved the maximum score for the problem on the example and outputting the majority label (3. If no such program was synthesized, our verifier instead uses transduction to predict the output labels of the test examples. Given the training examples as in-context examples, the VLM uses Chain of Thought (CoT) reasoning Wei et al. (2022) to label each held-out example as either positive or negative.

## 5 RESULTS

### 5.1 BASELINES

We evaluate GPT 4o and Claude 3.7 Sonnet on all three tasks. Full results are available in Table 1. For the solution task, two human raters judged whether model outputs were correct or incorrect, with no partial credit awarded. A rule is counted as correct only if both raters agreed on its correctness.

| Model | Task | | |
|---|---|---|---|
| | Verification | Solution (BP #2 - #100) | Inversion |
| Human Average | - | 47 | - |
| GPT-4o | 0.775 | 24 | 0.771 |
| GPT-4o + programs | 0.727 | - | - |
| GPT-4o + both | **0.79** | **31** | - |
| Claude 3.7 Sonnet | 0.835 | 44 | 0.838 |
| Claude 3.7 Sonnet + programs | 0.821 | - | - |
| Claude 3.7 Sonnet + both | **0.865** | **51** | - |

Table 1: Overall performance on the different tasks. Human data is taken from Wüst et al. (2025). "Verification" is classifying images given the ground truth label, "Solution" is outputting the correct natural language rule, and "Inversion" is the verification problem with positive and negative examples are swapped. For consistency with human data, only solution results for BPs #2-#100 are reported. Full data for all 100 problems can be found in Appendix A. We report accuracy for verification and inversion tasks and # problems solved for the solution task.

We find that Claude 3.7's performance is stronger on all three tasks. This is especially evident on the solution task, where Claude's accuracy is just below the average human performance on the BPs reported by Wüst et al. (2025) (47 correct), while 4o does not even solve a third of BPs.

## 5.2 VERIFICATION RESULTS

We experiment with both Claude and GPT as the VLM for both the program induction and transduction components of our verifier. Table 1 compares the performance of our method with a version of the verifier that ablates the natural language component, as well as with the performance of the base VLMs. For all results in Table 1, the number of programs synthesized per problem was 10.

The Claude model is able to achieve similar performance on the task with both programmatic verification only and the vanilla CoT model. However, as highlighted in Table 2, which analyzes the average performance of each verifier on different categories of BP problems (as introduced in Wüst et al. (2025)), these two methods differ in the types of problems they are able to solve. Programmatic verification appears to be advantageous for problems dealing with similarity between objects and with spatial concepts. This is unsurprising given the way in which spatial reasoning problems lend themselves to geometric or mathematical reasoning problems that can be encoded as formal programs with relative ease. Similarly, it does not come as a surprise that CoT outperforms programmatic verification on tasks requiring high-level conceptual knowledge; the gap between the two methods on number problems appears to be due to the fact that many BP number problems require each image to be broken down into a unique set of subparts that may be difficult to encode as a program. The strong results of both program verification and CoT, along with the complementary nature of the different problem categories they solve, lead to our combined method achieving improved results in nearly every BP category.

On the other hand, program verification with GPT-4o underperforms CoT by a decent margin, which may reflect the model's more limited capabilities in program synthesis when compared with Claude. The complementary nature of problems solved is not as evident as with Claude, and so the benefits of a combined method are more limited. This indicates that the effectiveness of combining programmatic and CoT reasoning is partially dependent on the model's abilities at these individual tasks.

| Model | Problem Category | | | | |
|---|---|---|---|---|---|
| | Concept | Number | Same | Size | Spatial |
| GPT-4o | **0.867** | 0.794 | **0.821** | 0.931 | **0.7** |
| GPT-4o + programs | 0.773 | 0.689 | 0.810 | 0.875 | 0.667 |
| GPT-4o + both | 0.846 | **0.833** | 0.702 | **0.958** | 0.681 |
| Claude 3.7 Sonnet | 0.904 | **0.878** | 0.845 | **0.972** | 0.744 |
| Claude 3.7 Sonnet + programs | 0.858 | 0.756 | **0.893** | 0.931 | 0.792 |
| Claude 3.7 Sonnet + both | **0.919** | 0.833 | **0.893** | 0.958 | **0.815** |

Table 2: Performance across BP categories from Wüst et al. (2025). + "programs" indicates that our method for verification with programs was used, while + "both" indicates that both natural language and programs were used by the verifier.

## 5.3 SOLUTION RESULTS

We test our full BP solving system with each of Claude and GPT as hypothesis generators. For the verification component of our solver, we utilize the verifier described above with slight modifications (see Appendix C. With Claude as the VLM, our system improves to **51** problems solved, exceeding the average human accuracy reported by Wüst et al. (2025). Utilizing our method also leads to improvement in 4o's score, although it remains far below human accuracy.

### 5.3.1 COMPARISON WITH HUMAN DATA

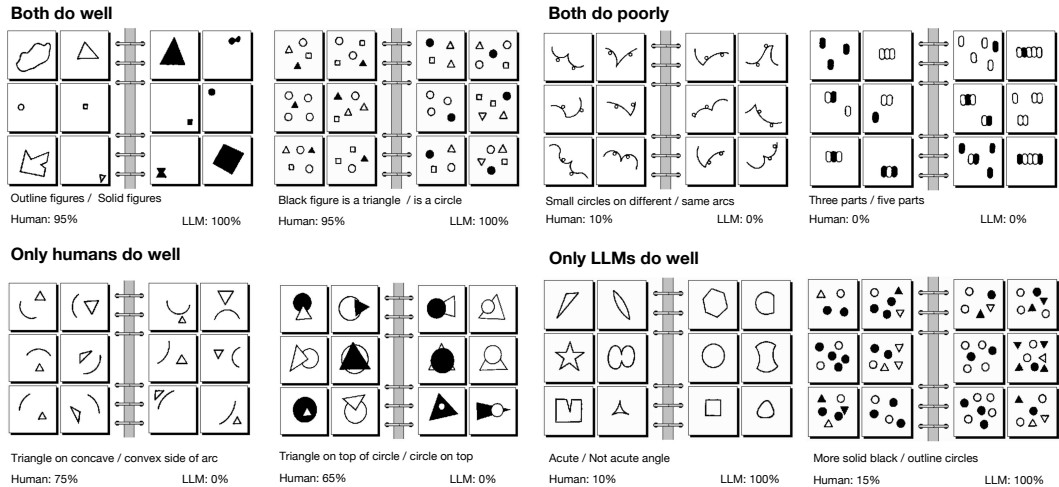

Figure 5: Examples of problems with similar and dissimilar performance

While reasoning over natural language and programs appears to be a promising means of improving VLM performance on reasoning problems like the BPs, whether it effectively models how humans solve these problems is a separate question. Figure 5 examines several examples of where human and model performance shows strong agreement or disagreement. The BPs that both humans and our system perform excep-

tionally well on tend to be conceptually simple, while the problems they both fail on either test particularly unusual combinations of visual concepts (BP #44, 'small circles on different arcs') or present a very simple ground truth concept in an intentionally complicated manner (BP #90, 'three' vs. 'five'). The problems where humans succeed and our system fails tend to either involve images that seem to be perceptually difficult for the VLM (BP #46, 'triangle on top of circle', where depth perception is important), or problems that compose a few concepts the VLM can independently recognize. BP #75 ('triangle on concave side of arc') is one such example, as the average VLM performance on earlier problems involving concavity (BP #4) and triangles (BP#6, #10) was similar to or exceeded that of humans. For this BP, the failure occurred because the VLM hypothesis generator was unable to generate the correct concept, highlighting the difficulty that VLMs still have with generalization and flexibility in utilizing concepts they can recognize independently.

### 5.3.2 MEMORIZATION

The strong out-of-the-box performance of the two VLMs (particularly the Claude model) raises questions about the extent to which the solutions to the BPs have been memorized. The BPs were published long before the advent of VLMs, meaning they are more than likely in the models' pretraining data.

To check for memorization, we invert the verification task: the positive concept and associated images are now negative examples, and likewise for the negative concept. We find that this perturbation of the task has very little impact on model performance (see Table 1), indicating that memorization may not be the primary driver of the observed performance of the VLMs.

### 5.4 DISCUSSION

**Natural language and code as complementary reasoning approaches.** Our analysis of the BP categories solved by natural language and code revealed that the shortcoming of one method was often the strength of the other. In particular, reasoning over natural language appears to be advantageous when high-level conceptual thinking is required, while reasoning over formal programs is preferable for reasoning problems involving exact calculations and fine-grained analysis. For VLMs that already excel in code generation, programmatic reasoning may be the key to shoring up weaknesses in ordinary CoT reasoning.

**Optimizable programmatic solutions to visual problems**. Allowing the programs we synthesized to be parameterizable and optimizable was central to our attempt to solve the BPs. Rather than attempt to synthesize the exact value of parameters along with the rest of the code, with our method the VLM can simply produce a high-level program and allow optimization to pinpoint exact values.

**Formalized Rules and Interpretability** Prior work such as Wüst et al. (2025) has largely focused on whether models can articulate abstract rules in natural language and then apply them. While this is informative, it leaves open the question of whether the stated rule actually governs the model's predictions. A model might claim to separate "spirals from non-spirals," yet rely on spurious cues in practice. By contrast, formalizing rules as executable programs provides a direct and verifiable link between reasoning and outcome. Programs can be run deterministically on held-out examples, making it clear whether a hypothesized rule truly accounts for correct classifications. Their structure also exposes intermediate operations that reveal how a solution is implemented and make failures easier to diagnose.

### 5.5 CONCLUSION

Formal programs and natural language are two distinct mediums for representing and reasoning about concepts, each with their own strengths and weaknesses with respect to flexibility, expressiveness, and precision. Through our investigation of the BPs we found that combining the two allows us to reason more accurately about the visual concepts central to the puzzles, allowing us to exceed average human accuracy for the first

time. Our hope is that this system, which is able to propose novel concepts and verify them adequately, is a first step toward AI systems that can acquire and use new concepts as flexibly as humans.

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

# A  OUTPUT RULES FOR SOLUTION TASK

The results in this section include BP#1, which was omitted above for consistency with human data.

## A.1  CLAUDE 3.7+PROGRAMS+COT SOLUTION RESULTS

| Problem Index | Rule | Rater 1 | Rater 2 | Final Score |
|---|---|---|---|---|
| 1 | Image is completely blank or white | 1 | 1 | 1 |
| 2 | The shapes in positive examples are larger, taking up a significant portion of the frame | 1 | 1 | 1 |
| 3 | No solid filled-in areas appear in the image | 1 | 1 | 1 |
| 4 | Negative examples all have at least one concave section (an indentation or inward curve in the shape's outline) | 1 | 1 | 1 |
| 5 | The shape is a polygon (has straight sides and vertices/corners) | 1 | 1 | 1 |
| 6 | The shape is a triangle or has a triangular element with a vertex pointing in the vertical direction (up or down) | 0 | 1 | 0 |
| 7 | Shapes have vertical orientation (taller than they are wide) | 1 | 1 | 1 |
| 8 | The shape is positioned on the right side of the image | 1 | 1 | 1 |
| 9 | Shapes with continuous curves or straight line segments only (no jagged edges) | 1 | 1 | 1 |
| 10 | The shape is triangular or has a triangular outline | 1 | 1 | 1 |
| 11 | Shapes that are elongated (longer in one dimension than the other) | 1 | 1 | 1 |
| 12 | Figures with a single primary axis of elongation (longer in one direction than others) | 1 | 1 | 1 |
| 13 | The shape is either a vertical rectangle or a horizontal ellipse | 1 | 1 | 1 |
| 14 | Images contain connected, continuous shapes (without disconnected elements) | 0 | 0 | 0 |
| 15 | The shape forms a continuous boundary that completely encloses an area | 1 | 1 | 1 |
| 16 | The pattern spirals or curves in a counter-clockwise direction when traced from outside to inside | 1 | 1 | 1 |

*Continued on next page*

| Problem Index | Rule | Rater 1 | Rater 2 | Final Score |
|---|---|---|---|---|
| 17 | The shape contains at least one "V" or wedge-like feature pointing inward or creating a notch in the shape | 1 | 1 | 1 |
| 18 | Shapes contain at least one area where the outline pinches inward from both sides | 1 | 1 | 1 |
| 19 | The shape has roughly balanced left and right sides connected by one or more horizontal segments in the middle, creating a shape that can be traversed left-to-right-to-left without lifting your pen | 0 | 1 | 0 |
| 20 | In positive examples, if you trace the curve from one dot to the other, both dots are placed along the outer perimeter of the shape | 0 | 0 | 0 |
| 21 | The largest shapes don't occur in fixed pairs (like large circle with large triangle) but rather have more variety in their sizes and relative positioning | 0 | 0 | 0 |
| 22 | Each positive image contains either multiple circles or a combination of different geometric shapes without any shape appearing to be significantly larger than others in the same image | 1 | 0 | 0 |
| 23 | Each positive image contains exactly one shape | 1 | 1 | 1 |
| 24 | The image contains at least one circle | 1 | 1 | 1 |
| 25 | The image contains at least one hollow circle (ring) and no solid/filled circles | 1 | 1 | 1 |
| 26 | Each positive image contains at least one solid shape (filled triangle or filled circle) | 0 | 0 | 0 |
| 27 | Each positive example contains at least two different types of filled shapes (among triangle, circle, and square) | 0 | 0 | 0 |
| 28 | The number of hollow circles (rings) does not exceed the number of solid circles in the image | 1 | 1 | 1 |
| 29 | In positive examples, there is at least one circle outside a closed shape, but the majority of circles are inside a closed shape | 1 | 0 | 0 |
| 30 | The image contains at least one intersection where lines or curves cross each other | 1 | 1 | 1 |

*Continued on next page*

| Problem Index | Rule | Rater 1 | Rater 2 | Final Score |
|---|---|---|---|---|
| 31 | In positive images, when there are crossings/intersections, they are self-intersections of the same curve, while negative images either have multiple disconnected components or intersections between different components | 1 | 1 | 1 |
| 32 | The shape has at least one sharp outward-pointing angle or corner | 1 | 1 | 1 |
| 33 | The shape contains at least one sharp point, acute angle, or cusp | 1 | 1 | 1 |
| 34 | The white shape is relatively large compared to its containing black shape (taking up significant interior space rather than being a small element) | 1 | 1 | 1 |
| 35 | The white shape/hole is aligned parallel to the major axis of the black shape | 1 | 1 | 1 |
| 36 | The triangle is positioned above the circle | 1 | 1 | 1 |
| 37 | The triangle is positioned above the circle | 1 | 1 | 1 |
| 38 | The triangle is larger than or equal to the circle in size | 1 | 1 | 1 |
| 39 | All line segments in the image are parallel or nearly parallel to each other | 1 | 1 | 1 |
| 40 | The image does not contain exactly 4 black circles | 0 | 0 | 0 |
| 41 | In positive examples, the solid black circles and hollow circles form alternating patterns where they are distributed in opposite sides or regions of the image | 0 | 0 | 0 |
| 42 | In positive examples, there is exactly one dot outside the enclosed shape | 0 | 0 | 0 |
| 43 | Each figure has peaks or heights that increase in size from left to right | 1 | 1 | 1 |
| 44 | All circles in the image are attached to the outer/convex side of curves rather than being enclosed by or on the inside/concave side of curves | 0 | 0 | 0 |
| 45 | In positive examples, when two shapes appear, the black shape is visually "following" or positioned after the white/outlined shape (when read left-to-right or top-to-bottom) | 0 | 0 | 0 |

*Continued on next page*

| Problem Index | Rule | Rater 1 | Rater 2 | Final Score |
|---|---|---|---|---|
| 46 | The triangle points toward or into the circle | 0 | 0 | 0 |
| 47 | If triangles and circles appear together in a nested relationship, the triangle must be inside the circle (not the circle inside the triangle) | 1 | 1 | 1 |
| 48 | Each positive image contains at least one hollow/outline shape (empty square or empty circle) AND at least one solid/filled shape (filled triangle, filled square, or filled circle) | 0 | 0 | 0 |
| 49 | Each positive image has exactly 3 small circles grouped closely together inside a larger shape | 1 | 1 | 1 |
| 50 | Positive examples have symmetrical arrangements of shapes, with elements balanced on the left and right sides of a vertical axis | 1 | 1 | 1 |
| 51 | The circles are arranged in a line pattern or can be connected to form straight lines | 0 | 0 | 0 |
| 52 | The direction of the arrows follows the curve they're on (arrows point along the path's direction rather than against it) | 0 | 0 | 0 |
| 53 | The outer shape has 4 or more sides, and the inner shape has the same or fewer sides than the outer shape | 1 | 1 | 1 |
| 54 | In positive examples, the three shapes (triangle, circle, and plus sign) always appear in a counter-clockwise arrangement when viewed from the center of the image | 1 | 1 | 1 |
| 55 | In positive examples, all small circles (loop shapes) are attached to the outside/edge of the main shape | 0 | 0 | 0 |
| 56 | All positive examples have shapes arranged in a way that no shape appears at the bottom center of the image | 0 | 0 | 0 |
| 57 | Each image contains exactly two identical shapes (same shape, same size, same style) | 1 | 1 | 1 |
| 58 | The image contains an even number of solid black squares (2 or 4) | 0 | 0 | 0 |

*Continued on next page*

| Problem Index | Rule | Rater 1 | Rater 2 | Final Score |
|---|---|---|---|---|
| 59 | The positive examples all contain two similar shapes (same shape category) of different sizes, while negative examples either contain different shape types or have shapes that aren't clearly related by similarity | 1 | 1 | 1 |
| 60 | Each image contains exactly three geometric shapes | 0 | 0 | 0 |
| 61 | In positive examples, the plus symbols appear on both sides of the line(s), with at least one plus symbol on each side of every line. | 0 | 0 | 0 |
| 62 | In positive examples, any line segment or curve never intersects/crosses itself or another line segment | 0 | 0 | 0 |
| 63 | All positive examples have shapes with the darker/shaded portion consistently on the right side or right edge of the figure | 1 | 1 | 1 |
| 64 | In positive examples, the small circle and the cross/plus sign are on opposite sides of the oval (one is above/below the oval while the other is to the left/right) | 0 | 0 | 0 |
| 65 | Circles surround or encircle the triangles | 0 | 0 | 0 |
| 66 | In positive examples, unconnected circles always appear in a straight horizontal line across the middle of the diagram | 1 | 1 | 1 |
| 67 | All branches curve in the same general direction (either all curve left or all curve right) within each positive example | 0 | 0 | 0 |
| 68 | The rightmost branch or curve starts higher on the vertical line than the leftmost branch or curve | 1 | 0 | 0 |
| 69 | The black circle (node) is positioned at the top of a vertical stem/line that connects directly to other curved branches | 1 | 1 | 1 |
| 70 | The main branches extend outward horizontally or diagonally from the trunk rather than growing primarily upward | 0 | 0 | 0 |
| 71 | Each positive image contains at least one circle with something inside it, or a larger shape containing a circle | 0 | 0 | 0 |

| Problem Index | Rule | Rater 1 | Rater 2 | Final Score |
|---|---|---|---|---|
| 72 | Positive examples have at least one endpoint that points in the same direction as another part of the line (either continuing in the same orientation or being parallel), while negative examples have endpoints pointing in different/diverging directions | 1 | 1 | 1 |
| 73 | Each image contains exactly three shapes: one ellipse/oval, one triangle, and one rectangle/quadrilateral | 0 | 0 | 0 |
| 74 | If the shape has a stem/stalk, it connects to the rounded end, not the pointed end | 1 | 1 | 1 |
| 75 | The curved line opens away from the triangle (the concave side of the arc faces away from the triangle) | 1 | 0 | 0 |
| 76 | Shapes that narrow in the middle like an hourglass or bow-tie | 1 | 1 | 1 |
| 77 | All angles in the figure are acute (less than 90 degrees) | 0 | 0 | 0 |
| 78 | The arrangement of lines is asymmetrical and appears random rather than structured | 0 | 0 | 0 |
| 79 | The triangle is never positioned above any circle | 0 | 0 | 0 |
| 80 | The circles and plus sign form a triangle pattern (not a straight line arrangement) | 0 | 0 | 0 |
| 81 | There is a clear spatial separation between filled objects and outlined objects, with no intermingling of the two types | 0 | 1 | 0 |
| 82 | The circle is positioned toward the center of the arrangement rather than at the edge | 0 | 0 | 0 |
| 83 | The circle is positioned in the center of the arrangement | 0 | 1 | 0 |
| 84 | The square is positioned outside or at the edge of the arrangement of circles, not in the center of the arrangement | 0 | 1 | 0 |
| 85 | Contains at most 3 distinct line segments total (where a line segment is a straight line without bends) | 1 | 1 | 1 |
| 86 | Negative examples all contain at least one point where 4 or more lines meet or intersect | 1 | 1 | 1 |

| Problem Index | Rule | Rater 1 | Rater 2 | Final Score |
|---|---|---|---|---|
| 87 | All angles in the figure are either right angles (90°) or diagonal angles (around 45° or 135°) | 0 | 0 | 0 |
| 88 | The arrangement must maintain symmetry or balance in the overall design | 0 | 0 | 0 |
| 89 | In each positive example, whenever two or more capsules are connected/touching, at least one of those connected capsules is filled (solid black) | 0 | 0 | 0 |
| 90 | All filled (black) shapes in the positive examples are symmetrically arranged from the center or edges | 0 | 0 | 0 |
| 91 | Each image contains elements that form specific geometric patterns with connected components (like branching structures, grouped shapes, or connected nodes) rather than disconnected or crossing elements | 0 | 0 | 0 |
| 92 | The pattern forms a closed or nearly closed loop (circle or oval shape) | 0 | 0 | 0 |
| 93 | The hollow circles form paths that diverge or split in multiple directions rather than following a single continuous curve | 0 | 0 | 0 |
| 94 | The black filled circle serves as a central connector or internal element in the chain of hollow circles | 0 | 1 | 0 |
| 95 | The internal lines are oriented vertically from top to bottom rather than horizontally from side to side | 1 | 1 | 1 |
| 96 | The pattern forms a triangular/pyramid shape (whether upright or inverted) with lines decreasing in length from center outward, or with lines arranged to form a triangular shape | 1 | 1 | 1 |
| 97 | The image contains a triangular shape or triangular arrangement of elements | 1 | 1 | 1 |
| 98 | Contains a triangle shape | 1 | 1 | 1 |
| 99 | Triangles and circles are intermixed/integrated throughout the image | 1 | 1 | 1 |
| 100 | The image contains the letter "A" (Latin letter A) | 1 | 1 | 1 |
| **Total** | | **56** | **58** | **52** |

## A.2  GPT-4O+PROGRAMS+COT SOLUTION RESULTS

| Problem Index | Rule | Rater 1 | Rater 2 | Final Score |
|---|---|---|---|---|
| 1 | Positive images are completely blank, without any shapes or marks. | 1 | 1 | 1 |
| 2 | The shape significantly fills the frame | 1 | 1 | 1 |
| 3 | contains no black-filled regions | 1 | 1 | 1 |
| 4 | Contains no irregular or concave edges | 1 | 1 | 1 |
| 5 | Contains shapes with only straight edges | 1 | 1 | 1 |
| 6 | Contains a triangle | 1 | 1 | 1 |
| 7 | shape is vertically elongated | 1 | 1 | 1 |
| 8 | The figure must have a convex polygonal property OR resemble basic geometric shapes | 0 | 0 | 0 |
| 9 | Contains smooth, continuous boundary with no sharp angles | 1 | 1 | 1 |
| 10 | Shape not a quadrilateral | 1 | 1 | 1 |
| 11 | The figure or one of its holes is non-regular or non-symmetrical | 0 | 0 | 0 |
| 12 | the figure is elongated or narrow | 1 | 1 | 1 |
| 13 | Shape is a vertically aligned rectangle or horizontally aligned ellipse | 1 | 0 | 0 |
| 14 | Contains smooth, connected curves or loops | 0 | 0 | 0 |
| 15 | Figure is fully enclosed without breaks | 0 | 1 | 0 |
| 16 | Contains a single spiral or looping element | 0 | 0 | 0 |
| 17 | Contains at least one sharp angle (less than 90°) | 0 | 0 | 0 |
| 18 | Contains a sharp inward-pointing angle | 0 | 0 | 0 |
| 19 | Contains at least one curve narrowing into a "neck" | 0 | 0 | 0 |
| 20 | The image contains at least one loop with a small circle attached to its outline. | 0 | 0 | 0 |
| 21 | Images must have at least one triangle or dot in an implied positional relationship. | 0 | 0 | 0 |

*Continued on next page*

| Problem Index | Rule | Rater 1 | Rater 2 | Final Score |
|---|---|---|---|---|
| 22 | An image is positive if all shapes are open or unfilled (e.g., no filled-in shapes). | 0 | 0 | 0 |
| 23 | Contains exactly one shape | 1 | 1 | 1 |
| 24 | The image contains at least one circle | 1 | 1 | 1 |
| 25 | No filled black circle is present | 1 | 1 | 1 |
| 26 | The only filled figures in the image are triangles | 0 | 0 | 0 |
| 27 | contains at least one solid black circle | 0 | 0 | 0 |
| 28 | Has more filled circles than unfilled circles | 1 | 1 | 1 |
| 29 | All circles are contained within a boundary. | 0 | 0 | 0 |
| 30 | The image contains intersecting curves or lines forming enclosed regions. | 1 | 1 | 1 |
| 31 | Contains no disconnected shapes | 0 | 0 | 0 |
| 32 | Must contain at least one sharp (non-smooth) angle | 1 | 0 | 0 |
| 33 | Contains at least one sharp angle or pointed projection | 1 | 1 | 1 |
| 34 | Contains a large, hollow region in the center | 1 | 1 | 1 |
| 35 | contains exactly one central hole | 0 | 0 | 0 |
| 36 | No circle appears above a triangle in the image. | 1 | 1 | 1 |
| 37 | The triangle is directly above the circle | 0 | 1 | 0 |
| 38 | Image is positive if the triangle in the image is larger than the circle. | 1 | 1 | 1 |
| 39 | All black lines in the image are parallel or approximately parallel | 1 | 1 | 1 |
| 40 | Dots do not form implied parallel lines | 0 | 0 | 0 |
| 41 | Contains three collinear points | 0 | 0 | 0 |
| 42 | At least three black points are evenly spaced with similar distances | 0 | 1 | 0 |
| 43 | At least one smooth curve is present | 0 | 0 | 0 |

| Problem Index | Rule | Rater 1 | Rater 2 | Final Score |
|---|---|---|---|---|
| 44 | Each small circle is located at the end of a curve. | 0 | 0 | 0 |
| 45 | At least one white shape is overlaid on a black shape | 1 | 1 | 1 |
| 46 | Solid (black-filled) circles are present | 0 | 0 | 0 |
| 47 | No filled circle inside any triangle | 0 | 0 | 0 |
| 48 | Only filled shapes are on top | 1 | 1 | 1 |
| 49 | A group of 4 or more dots must be enclosed within a single outlined shape. | 0 | 0 | 0 |
| 50 | Contains shapes arranged in symmetry about a center point | 0 | 1 | 0 |
| 51 | Points are not evenly spaced | 0 | 0 | 0 |
| 52 | curve is continuous and does not self-intersect | 0 | 0 | 0 |
| 53 | The inner shape has fewer sides than the outer shape | 1 | 1 | 1 |
| 54 | No symbol is isolated in the bottom-left corner. | 0 | 0 | 0 |
| 55 | The small circle is attached to the boundary of the shape rather than floating away | 0 | 0 | 0 |
| 56 | No mixture of filled and outlined shapes | 1 | 1 | 1 |
| 57 | All shapes in the image are the same size. | 0 | 1 | 0 |
| 58 | contains exactly one hollow square | 0 | 0 | 0 |
| 59 | All objects are the same shape (e.g., both circles, both triangles). | 0 | 1 | 0 |
| 60 | If the image contains two or more identical shapes. | 0 | 1 | 0 |
| 61 | A "+" symbol exists on both sides of any adjacent line | 0 | 0 | 0 |
| 62 | Contains smooth curves or wavy lines without sharp edges | 0 | 0 | 0 |
| 63 | Oval is oriented diagonally (not perfectly horizontal or vertical) | 0 | 0 | 0 |

| Problem Index | Rule | Rater 1 | Rater 2 | Final Score |
|---|---|---|---|---|
| 64 | Shapes are distributed randomly, not stacked or lined up | 0 | 0 | 0 |
| 65 | Triangles and circles are evenly distributed (not overly grouped in one main cluster) | 0 | 0 | 0 |
| 66 | Extra circular nodes are not forming vertical groups | 1 | 1 | 1 |
| 67 | The image contains a vertical line with two symmetrical curves. | 0 | 0 | 0 |
| 68 | The left branch begins at a lower vertical position than the right branch. | 1 | 0 | 0 |
| 69 | Circle is attached to a line | 0 | 0 | 0 |
| 70 | Contains lines with fewer crossings or simpler branching | 0 | 1 | 0 |
| 71 | The image contains exactly one nested enclosure | 0 | 0 | 0 |
| 72 | One continuous curve with no sharp angles | 0 | 0 | 0 |
| 73 | All three shapes in the image differ in orientation. | 0 | 0 | 0 |
| 74 | The shape must have at least one concave side | 0 | 0 | 0 |
| 75 | The curved shape has an open end facing outward (not closed around another feature). | 0 | 0 | 0 |
| 76 | Shape does not have a closed loop | 0 | 0 | 0 |
| 77 | The right branch begins higher than the left branch. | 0 | 0 | 0 |
| 78 | Contains exactly three lines, all with differing orientations | 0 | 0 | 0 |
| 79 | contains a black circle and a white circle that are parallel | 0 | 0 | 0 |
| 80 | Exactly two filled circles are present in the image. | 0 | 0 | 0 |
| 81 | Images containing at least one pair of identical shapes (based on size and fill) are positive. | 0 | 0 | 0 |

*Continued on next page*

| Problem Index | Rule | Rater 1 | Rater 2 | Final Score |
|---|---|---|---|---|
| 82 | There is exactly one outlined circle. | 0 | 0 | 0 |
| 83 | The convex hull of crosses always overlaps the convex hull of the circle. | 1 | 1 | 1 |
| 84 | The small square is near the exterior boundary of the arrangement | 0 | 1 | 0 |
| 85 | geometric configuration formed by no more than 3 objects | 1 | 0 | 0 |
| 86 | The image contains exactly one or two vertices where three lines meet. | 1 | 1 | 1 |
| 87 | Contains no intersections of lines | 0 | 0 | 0 |
| 88 | Each cap structure must exactly enclose one other cap | 0 | 0 | 0 |
| 89 | contains three or fewer total shapes | 1 | 0 | 0 |
| 90 | Each row contains an odd number of grouped oval shapes. | 0 | 0 | 0 |
| 91 | Contains exactly three components | 1 | 1 | 1 |
| 92 | Contains a continuous curve with at least one large closed or rounded section | 0 | 1 | 0 |
| 93 | The image contains exactly three distinct groups of dots. | 0 | 0 | 0 |
| 94 | The black circle (if present) is never at the endpoint of the structure | 1 | 1 | 1 |
| 95 | Contains parallel vertical lines | 0 | 1 | 0 |
| 96 | The lines in positive examples form patterns that converge (e.g., triangles or peaks), whereas lines in negative examples are parallel or form a uniform grid. | 0 | 0 | 0 |
| 97 | The design emphasizes a triangular pattern, not circular patterns. | 1 | 1 | 1 |
| 98 | The image contains at least one triangle. | 1 | 1 | 1 |
| 99 | Triangles and circles are mixed together in the pattern. | 1 | 1 | 1 |
| 100 | No loops or circles in the figure | 0 | 1 | 0 |
| **Total** | | **37** | **44** | **32** |

## A.3 CLAUDE 3.7 (BASELINE) SOLUTION RESULTS

| Problem Index | Rule | Rater 1 | Rater 2 | Score |
|---|---|---|---|---|
| 1 | An image is positive if it is blank/empty with no visible content, while an image is negative if it contains any visible shapes, patterns, or lines | 1 | 1 | 1 |
| 2 | whether the shape extends to/near the edges of the frame (positive) or is contained well within the boundaries with significant white space around it (negative) | 1 | 0 | 0 |
| 3 | An image is positive if it shows only the outline/border of a shape (hollow), and negative if it shows a solid/filled shape | 1 | 1 | 1 |
| 4 | A shape is positive if and only if it is convex (has no indentations or inward curves), while a shape is negative if it has at least one concave section | 1 | 1 | 1 |
| 5 | Positive images contain only shapes with straight edges and vertices, while negative images contain only shapes with curved edges and no vertices | 1 | 1 | 1 |
| 6 | An image is positive if and only if it contains a shape with exactly 3 sides (a triangle), while an image is negative if it contains a shape with exactly 4 sides (a quadrilateral) | 1 | 1 | 1 |
| 7 | An image is positive if the shape has a vertical orientation (taller than wide), and negative if the shape has a horizontal orientation (wider than tall) | 1 | 1 | 1 |
| 8 | the position of the shape within the frame: positive examples have shapes on the right side, while negative examples have shapes on the left side | 1 | 1 | 1 |
| 9 | Positive images have smooth outlines (either straight lines or curves), while negative images have jagged, zigzag outlines with many small angular segments | 1 | 1 | 1 |

*Continued on next page*

| Problem Index | Rule | Rater 1 | Rater 2 | Score |
|---|---|---|---|---|
| 10 | that positive images contain triangular shapes (3-sided polygons) while negative images contain quadrilateral shapes (4-sided polygons) | 1 | 1 | 1 |
| 11 | that positive images contain shapes with low rotational symmetry (0 or 2-fold), while negative images contain shapes with higher rotational symmetry (3-fold or greater) | 0 | 1 | 0 |
| 12 | Positive images contain shapes that have a single clear axis of elongation, while negative images contain shapes that either have equal dimensions in multiple directions or have multiple axes of similar length/importance | 1 | 1 | 1 |
| 13 | An image is positive if and only if the shape's major axis aligns with its natural orientation (vertical for rectangles, horizontal for ellipses), while an image is negative if the shape's major axis contradicts its natural orientation | 1 | 1 | 1 |
| 14 | Positive examples contain at least one shape or structure that creates a boundary or enclosure (either complete or nearly complete), while negative examples lack this property of enclosure | 0 | 0 | 0 |
| 15 | An image is positive if and only if it contains a completely closed shape (a continuous line that forms a complete loop with no gaps or openings), while negative images contain shapes with at least one opening or gap in their boundary | 1 | 1 | 1 |
| 16 | the direction of rotation or winding: counterclockwise rotation for positive examples and clockwise rotation for negative examples | 1 | 1 | 1 |
| 17 | An image is positive if and only if the shape contains at least one concave angle (an inward-pointing vertex where the interior angle is greater than 180 degrees) | 1 | 1 | 1 |

*Continued on next page*

| Problem Index | Rule | Rater 1 | Rater 2 | Score |
|---|---|---|---|---|
| 18 | Positive images contain at least one constriction point where the shape narrows significantly between wider sections, while negative images lack such pinch points and maintain more consistent width throughout their form | 1 | 1 | 1 |
| 19 | that positive examples have their main connecting element oriented horizontally (connecting left and right parts), while negative examples have their main connecting element oriented vertically (connecting top and bottom parts) | 1 | 1 | 1 |
| 20 | An image is positive if and only if a straight line connecting the two dots does not intersect the boundary of the shape (the dots have a direct "line of sight" to each other) | 0 | 0 | 0 |
| 21 | An image is positive if and only if it contains at least one small circle AND at least one triangle (of any size) | 0 | 0 | 0 |
| 22 | An image is positive if and only if the geometric shapes are arranged along a diagonal axis across the image | 0 | 0 | 0 |
| 23 | An image is positive if and only if it contains exactly one geometric shape | 1 | 1 | 1 |
| 24 | An image is positive if and only if it contains at least one circle | 1 | 1 | 1 |
| 25 | Positive images must contain both a hollow circle () AND a filled triangle (), while negative images contain at least one filled circle () and never have both a hollow circle and filled triangle together | 1 | 1 | 1 |
| 26 | An image is positive if and only if it contains BOTH at least one solid triangle AND at least one hollow circle within the same image | 1 | 1 | 1 |
| 27 | An image is positive if and only if it contains at least one solid/filled triangle | 0 | 0 | 0 |
| 28 | An image is positive if and only if the number of filled circles exceeds the number of hollow circles | 1 | 1 | 1 |

*Continued on next page*

| Problem Index | Rule | Rater 1 | Rater 2 | Score |
|---|---|---|---|---|
| 29 | An image is positive if and only if it contains exactly one circle outside the closed shape, while negative images contain multiple (two or more) circles outside the closed shape | 0 | 0 | 0 |
| 30 | An image is positive if and only if it contains at least one intersection where lines or curves cross over each other | 1 | 1 | 1 |
| 31 | An image is positive if and only if it consists of exactly one continuous curve (can be drawn without lifting the pen), while an image is negative if it contains two or more disconnected components that would require lifting the pen to draw the complete image | 1 | 1 | 1 |
| 32 | An image is positive if it contains at least one sharp, acute angle or pointed vertex; negative images lack such sharp points and are characterized by smoother, more rounded contours | 1 | 0 | 0 |
| 33 | Positive images contain at least one acute angle or sharp point in their geometry, while negative images lack any acute angles or sharp points | 1 | 1 | 1 |
| 34 | An image is positive if the white shape inside the black shape is large and extends to (or near) the edge of the black shape | 1 | 1 | 1 |
| 35 | An image is positive if and only if the white internal element is oriented parallel to the major axis of the containing shape | 1 | 1 | 1 |
| 36 | an image is positive if and only if the triangle is positioned in a higher or more rightward position relative to the circle | 0 | 0 | 0 |
| 37 | the directional arrangement of the three shapes: clockwise arrangement for positive images and counterclockwise arrangement for negative images | 0 | 0 | 0 |
| 38 | An image is positive if and only if the triangle is larger than or equal in size to the circle; an image is negative if the circle is larger than the triangle | 1 | 1 | 1 |

*Continued on next page*

| Problem Index | Rule | Rater 1 | Rater 2 | Score |
|---|---|---|---|---|
| 39 | An image is positive if and only if all three line segments are parallel to each other (have the same orientation) | 1 | 1 | 1 |
| 40 | An image is positive if the 4 dots cannot form a rectangle when connected, while an image is negative if the 4 dots can form a rectangle (including squares) when connected | 0 | 0 | 0 |
| 41 | An image is positive if and only if all solid circles can be separated from all hollow circles by a single straight line (linear separability) | 0 | 0 | 0 |
| 42 | An image is positive if and only if it contains a shape with exactly 3 dots inside arranged in a roughly linear pattern (where the dots can be connected by a relatively straight line) | 1 | 1 | 1 |
| 43 | An image is positive if the pattern shows increasing amplitude/height from left to right, and negative if the pattern shows decreasing amplitude/height from left to right | 1 | 1 | 1 |
| 44 | An image is positive if and only if all circles are positioned on the convex (outward) side of curves | 0 | 0 | 0 |
| 45 | an image is positive if the filled (black) shape appears to the right of or below the unfilled (white) shape; an image is negative if the filled (black) shape appears to the left of or above the unfilled (white) shape | 0 | 0 | 0 |
| 46 | An image is positive if and only if there is a directional "pointing" relationship between the shapes, where either a vertex of the triangle points toward the circle or a portion of the circle points toward the triangle | 0 | 0 | 0 |
| 47 | An image is positive if and only if it contains at least one triangle inside a circle | 1 | 1 | 1 |
| 48 | An image is positive if and only if it contains exactly one hollow square (outline square) | 0 | 0 | 0 |

*Continued on next page*

| Problem Index | Rule | Rater 1 | Rater 2 | Score |
|---|---|---|---|---|
| 49 | In positive images, the cluster of 3 small circles appears INSIDE the large geometric shape, while in negative images, the cluster of 3 small circles appears OUTSIDE the large geometric shape | 0 | 1 | 0 |
| 50 | An image is positive if and only if it has vertical (left-right) symmetry, meaning it can be divided by a vertical line into two mirror-image halves | 1 | 1 | 1 |
| 51 | An image is positive if and only if at least 3 of its circles can be connected to form a straight line (either perfectly horizontal, vertical, or diagonal) | 0 | 0 | 0 |
| 52 | whether all arrows in the image indicate the same rotational direction (positive) or opposite rotational directions (negative) | 0 | 0 | 0 |
| 53 | An image is positive if and only if the number of sides of the inner shape is less than or equal to the number of sides of the outer shape | 1 | 1 | 1 |
| 54 | the directional arrangement of the shapes: if you trace a path from triangle to circle to plus sign and back to triangle, positive examples follow a counterclockwise path, while negative examples follow a clockwise path | 1 | 1 | 1 |
| 55 | An image is positive if it contains a small circle that is attached to the right side of the main shape, and negative if the small circle is attached to the left side of the main shape | 0 | 0 | 0 |
| 56 | An image is positive if and only if it contains at least one triangle that is positioned vertically higher than at least one circle | 0 | 0 | 0 |
| 57 | An image is positive if and only if it contains exactly two identical shapes | 1 | 1 | 1 |
| 58 | An image is positive if the shapes are distributed to occupy at least three of the four quadrants (or corners) of the image | 0 | 0 | 0 |

| Problem Index | Rule | Rater 1 | Rater 2 | Score |
|---|---|---|---|---|
| 59 | An image is positive if it contains two shapes of the same type but different sizes, while an image is negative if it contains two shapes of different types | 1 | 1 | 1 |
| 60 | An image is positive if and only if it contains at least one shape that is positioned both to the right of some shape AND below some shape (not necessarily the same reference shape) | 0 | 0 | 0 |
| 61 | An image is positive if and only if the line divides the space such that there is at least one plus sign on each side of the line | 0 | 0 | 0 |
| 62 | Positive images have at least one endpoint that remains "free" (not connected back to the shape and not crossing itself), while negative images have shapes that either form closed loops, have self-intersections, or have all endpoints connecting back to form closed figures | 0 | 0 | 0 |
| 63 | Positive examples all have shapes with a distinct straight or thicker right edge, while negative examples lack this consistent right-side emphasis | 1 | 1 | 1 |
| 64 | that positive images have ellipses tilted at an angle, while negative images have vertically oriented ellipses | 0 | 0 | 0 |
| 65 | An image is positive if the circles are arranged to surround or encircle the triangles, creating a containment relationship | 0 | 0 | 0 |
| 66 | An image is positive if and only if it contains a horizontal row of disconnected circles that acts as a dividing line, separating the connected network into distinct regions above and below this row | 0 | 1 | 0 |
| 67 | An image is positive if the curved elements are on the left side of the structure, and negative if the curved elements are on the right side of the structure | 0 | 0 | 0 |

*Continued on next page*

| Problem Index | Rule | Rater 1 | Rater 2 | Score |
|---|---|---|---|---|
| 68 | An image is positive if all curved segments have their concavity facing left, while an image is negative if all curved segments have their concavity facing right | 0 | 0 | 0 |
| 69 | An image is positive if the curved branches open toward the right (concavity faces right), and negative if the curved branches open toward the left (concavity faces left) | 0 | 0 | 0 |
| 70 | that positive images show horizontally oriented branching structures with a curved main branch and asymmetrical growth pattern, while negative images show vertically oriented branching structures with more symmetrical, multi-directional growth patterns | 0 | 0 | 0 |
| 71 | An image is positive if and only if it contains at least one instance of a shape that is completely contained within another shape | 0 | 0 | 0 |
| 72 | An image is positive if and only if the curve has at least one horizontal endpoint (pointing left or right) positioned at the bottom or lower portion of the image | 0 | 0 | 0 |
| 73 | the directional arrangement of the three shapes: clockwise arrangement indicates a positive image, while counterclockwise arrangement indicates a negative image | 0 | 0 | 0 |
| 74 | An image is positive if and only if it contains at least one sharp point or corner in its shape | 0 | 0 | 0 |
| 75 | An image is positive if and only if it contains a curved line that opens toward the left side AND has a triangle positioned to the right of the curved line | 0 | 0 | 0 |
| 76 | An image is positive if the curves on opposite sides bend in opposite directions (one concave, one convex), creating a shape with a pinched middle | 0 | 1 | 0 |
| 77 | An image is positive if and only if all angles formed between any two adjacent lines at all vertices are acute (less than 90 degrees) | 0 | 0 | 0 |

*Continued on next page*

| Problem Index | Rule | Rater 1 | Rater 2 | Score |
|---|---|---|---|---|
| 78 | An image is positive if and only if no two lines in the image are parallel to each other | 0 | 0 | 0 |
| 79 | An image is positive if the three shapes form a triangular arrangement (non-collinear), while an image is negative if the three shapes are arranged in a more linear fashion (collinear or nearly collinear) | 0 | 0 | 0 |
| 80 | that in positive images, the plus sign and the two dots form a right angle (90-degree angle) when connected by imaginary lines, with the plus sign at the vertex of the angle | 0 | 0 | 0 |
| 81 | Positive images have a clear spatial separation between filled (solid) shapes and unfilled (outline) shapes, while negative images have these two types of shapes mixed together without a clear spatial division | 0 | 1 | 0 |
| 82 | An image is positive if the circle is positioned at the periphery/edge of the arrangement, away from the center and not surrounded by plus signs | 0 | 0 | 0 |
| 83 | An image is positive if and only if the circle is centrally located and surrounded by plus signs from multiple directions (at least 3-4 different sides), creating an enclosure pattern | 0 | 1 | 0 |
| 84 | An image is positive if the square is positioned outside or at the periphery of the circle arrangement; an image is negative if the square is positioned in the center or near the center of the circle arrangement | 1 | 1 | 1 |
| 85 | An image is positive if and only if it contains exactly 3 connected line segments | 0 | 0 | 0 |
| 86 | An image is positive if and only if every junction in the image has at most 3 line segments meeting at a point | 1 | 1 | 1 |
| 87 | An image is positive if and only if all angles formed by intersecting or connecting line segments are multiples of 45 degrees | 0 | 0 | 0 |

*Continued on next page*

| Problem Index | Rule | Rater 1 | Rater 2 | Score |
|---|---|---|---|---|
| 88 | An image is positive if and only if it contains at least one "sandwich" arrangement where either a filled black capsule is positioned between two hollow capsules, or a hollow capsule is positioned between two filled black capsules | 0 | 0 | 0 |
| 89 | An image is positive if and only if it contains at least one filled black capsule that is directly adjacent to (touching) at least one hollow capsule | 0 | 0 | 0 |
| 90 | An image is positive if and only if it contains an even number of filled (black) capsules that is greater than zero | 0 | 0 | 0 |
| 91 | An image is positive if and only if it contains exactly three primary elements or components | 1 | 1 | 1 |
| 92 | In positive examples, solid black circles mark the corners or turning points where the pattern changes direction, while in negative examples, solid circles (if present) are positioned randomly or along straight segments of the pattern | 0 | 0 | 0 |
| 93 | An image is positive if and only if no three solid black circles can be connected to form a straight line | 0 | 0 | 0 |
| 94 | An image is positive if the filled (black) circle is in a middle position serving as a junction point between branches, while an image is negative if the filled (black) circle is at an endpoint of the sequence | 0 | 1 | 0 |
| 95 | the orientation of the lines: positive images contain vertical lines (running top to bottom), while negative images contain horizontal lines (running left to right) | 1 | 1 | 1 |
| 96 | An image is positive if and only if the arrangement of lines forms a shape that points to the left ($\leftarrow$) | 0 | 0 | 0 |
| 97 | that positive images contain triangular shapes (3 sides/corners) while negative images contain circular shapes (continuous curvature with no corners) | 1 | 1 | 1 |

| Problem Index | Rule | Rater 1 | Rater 2 | Score |
|---|---|---|---|---|
| 98 | An image is positive if it contains a triangle (3-sided polygon) and negative if it contains a quadrilateral (4-sided polygon) | 1 | 1 | 1 |
| 99 | An image is positive if circles and triangles are randomly mixed together throughout the space; an image is negative if circles and triangles are spatially segregated into their own distinct regions | 1 | 1 | 1 |
| 100 | that positive images show the letter "A" while negative images show the letter "B" | 1 | 1 | 1 |
| **Total** | | **47** | **52** | **45** |

## A.4   GPT-4O (BASELINE) SOLUTION RESULTS

| Problem Index | Rule | Rater 1 | Rater 2 | Score |
|---|---|---|---|---|
| 1 | An image is positive if it contains a single, closed, smooth, and continuous shape with no sharp angles or straight edges. An image is negative if it contains sharp angles, straight edges, disconnected elements, or multiple distinct shapes. | 0 | 0 | 0 |
| 2 | An image is positive if it contains a large, centered shape that occupies a significant portion of the image. An image is negative if it contains a small, off-center shape that occupies only a small portion of the image. | 0 | 1 | 0 |
| 3 | - An image is positive if it contains hollow (outlined) shapes. - An image is negative if it contains filled (solid) shapes. | 1 | 1 | 1 |
| 4 | An image is positive if it is a closed shape that is either a smooth curve or a regular polygon with straight sides. An image is negative if it is irregular, wavy, or not a closed shape. | 0 | 1 | 0 |
| 5 | A shape is positive if it is a polygon (composed of straight lines and vertices). A shape is negative if it contains curves or lacks straight edges. | 1 | 1 | 1 |

| Problem Index | Rule | Rater 1 | Rater 2 | Score |
|---|---|---|---|---|
| 6 | - An image is positive if it contains a triangle (a three-sided polygon). - An image is negative if it does not contain a triangle. | 1 | 1 | 1 |
| 7 | An image is positive if the shape is vertically oriented (elongated along the vertical axis). An image is negative if the shape is horizontally oriented (elongated along the horizontal axis or spread horizontally). | 1 | 1 | 1 |
| 8 | An image is positive if the shape is upright, centered, and in its standard geometric form. An image is negative if the shape is rotated, flipped, distorted, or otherwise not in its standard orientation. | 0 | 0 | 0 |
| 9 | An image is positive if its edges are smooth and continuous, without jagged or spiky features. An image is negative if its edges are jagged, with sharp points or angular transitions. | 1 | 1 | 1 |
| 10 | An image is positive if the shape is a triangle or derived from a triangle. An image is negative if the shape is not a triangle. | 1 | 1 | 1 |
| 11 | An image is positive if it contains an open boundary, an irregular shape, or an inner cutout. An image is negative if it is a closed, regular, and symmetrical shape. | 0 | 0 | 0 |
| 12 | - An image is positive if it contains an open shape (a shape with at least one gap or opening). - An image is negative if it contains a closed shape (a shape with no gaps or openings). | 0 | 0 | 0 |
| 13 | An image is positive if the shape is vertically oriented (taller than it is wide). An image is negative if the shape is horizontally oriented (wider than it is tall). | 0 | 0 | 0 |
| 14 | An image is positive if it contains continuous, smooth, or connected shapes with flowing curves or cohesive structures. An image is negative if it contains disconnected, sharp, or angular elements or isolated shapes. | 0 | 0 | 0 |

| Problem Index | Rule | Rater 1 | Rater 2 | Score |
|---|---|---|---|---|
| 15 | An image is positive if it contains a closed shape with no breaks or self-intersections. An image is negative if the shape is open, self-intersecting, or has a gap in its outline. | 1 | 1 | 1 |
| 16 | An image is positive if it contains a spiral or nested shape that is incomplete or broken. An image is negative if it contains a spiral or nested shape that is complete and unbroken. | 0 | 0 | 0 |
| 17 | A shape is positive if it has at least one sharp point or angle and is not fully enclosed (i.e., it has an open boundary). A shape is negative if it is fully enclosed and lacks sharp points or angles. | 0 | 0 | 0 |
| 18 | An image is positive if the shape is symmetrical (either reflective or rotational). An image is negative if the shape is asymmetrical. | 0 | 0 | 0 |
| 19 | An image is positive if it contains a closed loop that is intersected or connected by a straight line. Otherwise, it is negative. | 0 | 0 | 0 |
| 20 | An image is positive if the curve forms a simple closed loop (does not cross itself) and encloses the two dots. An image is negative if the curve crosses itself at least once. | 0 | 0 | 0 |
| 21 | An image is positive if and only if it contains at least one non-inverted triangle and one circle. | 0 | 0 | 0 |
| 22 | An image is positive if it contains at least two shapes of the same type. An image is negative if all shapes in the image are of different types. | 0 | 0 | 0 |
| 23 | An image is positive if it contains exactly one shape. An image is negative if it contains two shapes. | 1 | 1 | 1 |
| 24 | An image is positive if it contains at least one circle. Otherwise, it is negative. | 1 | 1 | 1 |
| 25 | An image is positive if it contains a black triangle. Otherwise, it is negative. | 1 | 1 | 1 |

| Problem Index | Rule | Rater 1 | Rater 2 | Score |
|---|---|---|---|---|
| 26 | - An image is positive if it contains a mix of solid and hollow shapes. - An image is negative if all shapes are either all solid or all hollow. | 0 | 0 | 0 |
| 27 | - An image is positive if it contains at least one solid shape and at least one hollow shape. - An image is negative if it contains only solid shapes or only hollow shapes. | 0 | 0 | 0 |
| 28 | An image is positive if it contains at least one filled circle and one unfilled circle. Otherwise, it is negative. | 0 | 0 | 0 |
| 29 | - An image is **positive** if at least one smaller circle is **inside** a closed shape. - An image is **negative** if no smaller circles are **inside** a closed shape. | 0 | 0 | 0 |
| 30 | - An image is positive if it contains at least one intersection where lines or curves cross each other. - An image is negative if it does not contain any intersections. | 1 | 1 | 1 |
| 31 | An image is positive if it consists of a single continuous and closed curve without intersections or overlaps of separate curves. An image is negative if it contains multiple disconnected curves, open shapes, or intersections of separate curves. | 0 | 1 | 0 |
| 32 | An image is positive if it contains at least one sharp corner or pointed edge and is not fully enclosed. An image is negative if it is a fully enclosed shape with smooth, rounded edges. | 1 | 0 | 0 |
| 33 | An image is positive if it contains sharp angles, irregularity, or intersections. An image is negative if it is a regular geometric shape, smooth, or symmetrical. | 1 | 0 | 0 |
| 34 | An image is positive if the hole is not centered within the larger shape. An image is negative if the hole is centered within the larger shape. | 0 | 0 | 0 |

| Problem Index | Rule | Rater 1 | Rater 2 | Score |
|---|---|---|---|---|
| 35 | An image is positive if the hole is centered and aligned with the larger shape. An image is negative if the hole is off-center or misaligned with the larger shape. | 0 | 1 | 0 |
| 36 | An image is positive if the triangle is positioned above the circle. Otherwise, it is negative. | 1 | 1 | 1 |
| 37 | - An image is positive if the triangle is the topmost shape. Otherwise, the image is negative. | 0 | 0 | 0 |
| 38 | An image is positive if the circle is smaller than the triangle. An image is negative if the circle is larger than or equal in size to the triangle. | 1 | 1 | 1 |
| 39 | An image is positive if all lines are parallel to at least one other line. An image is negative if there is at least one line that is not parallel to any other line. | 1 | 1 | 1 |
| 40 | An image is positive if no three dots are collinear. An image is negative if at least three dots are collinear. | 1 | 0 | 0 |
| 41 | - An image is positive if no two filled (black) circles are adjacent. - An image is negative if at least two filled (black) circles are adjacent. | 0 | 0 | 0 |
| 42 | An image is positive if and only if there is a closed shape that encloses exactly three dots. If the number of enclosed dots is not three, the image is negative. | 0 | 0 | 0 |
| 43 | An image is positive if it has a consistent, alternating wave-like pattern of peaks and valleys. An image is negative if the pattern is irregular, includes vertical segments, or has abrupt changes in the wave-like flow. | 0 | 0 | 0 |
| 44 | - An image is positive if the dots are on the **outer side** of the curve. - An image is negative if the dots are on the **inner side** of the curve. | 0 | 0 | 0 |

| Problem Index | Rule | Rater 1 | Rater 2 | Score |
|---|---|---|---|---|
| 45 | An image is positive if the white shape is fully enclosed by the black shape, creating a "hole" or "cut-out" effect. An image is negative if the white shape is not fully enclosed by the black shape. | 0 | 0 | 0 |
| 46 | - An image is positive if the triangle partially overlaps the circle, and the triangle's interior is visible within the circle. - An image is negative if the circle is fully inside the triangle, the triangle is fully inside the circle, or the shapes touch without overlapping in the specified manner. | 0 | 1 | 0 |
| 47 | An image is positive if and only if it contains at least one triangle that fully encloses a circle. If no triangle fully encloses a circle, the image is negative. | 1 | 0 | 0 |
| 48 | - An image is positive if it contains at least one solid shape and at least one hollow shape. - An image is negative if it contains only solid shapes or only hollow shapes. | 0 | 0 | 0 |
| 49 | - An image is positive if the smaller circles inside the large geometric shape are clustered together, and there are additional smaller circles outside the shape. - An image is negative if the smaller circles outside the large geometric shape are clustered together, or the smaller circles inside the shape are not clustered. | 1 | 1 | 1 |
| 50 | An image is positive if it exhibits symmetry, balance, and alignment in the arrangement of its shapes. An image is negative if it lacks these qualities and appears disorganized or asymmetrical. | 0 | 1 | 0 |
| 51 | An image is positive if no three circles are collinear. An image is negative if three or more circles are collinear. | 0 | 0 | 0 |
| 52 | An image is positive if the two arrows point in opposite directions. An image is negative if the two arrows point in the same direction. | 0 | 1 | 0 |

*Continued on next page*

| Problem Index | Rule | Rater 1 | Rater 2 | Score |
|---|---|---|---|---|
| 53 | An image is positive if the inner shape is different from the outer shape. An image is negative if the inner shape is the same as the outer shape. | 0 | 0 | 0 |
| 54 | An image is positive if the triangle and the circle are on opposite sides of the plus sign. An image is negative if the triangle and the circle are on the same side of the plus sign. | 0 | 0 | 0 |
| 55 | - An image is positive if the small circle is connected to the closed shape via a smooth, curved connection. - An image is negative if the small circle is connected to the closed shape via a sharp/angular connection or a straight line. | 0 | 0 | 0 |
| 56 | - An image is positive if it does not contain any filled circles. - An image is negative if it contains at least one filled circle. | 0 | 0 | 0 |
| 57 | An image is positive if and only if the two shapes in the image are identical in type, size, and orientation. Otherwise, the image is negative. | 1 | 1 | 1 |
| 58 | An image is positive if and only if it contains at least one solid black square. Otherwise, it is negative. | 0 | 0 | 0 |
| 59 | - An image is positive if it contains two shapes that are identical in shape but differ in size. - An image is negative if the two shapes are not identical in shape. | 1 | 1 | 1 |
| 60 | An image is positive if it contains at least one pair of identical shapes. An image is negative if all shapes in the image are distinct. | 0 | 1 | 0 |
| 61 | - An image is positive if no "+" symbol is intersected by the line. - An image is negative if at least one "+" symbol is intersected by the line. | 0 | 0 | 0 |

*Continued on next page*

| Problem Index | Rule | Rater 1 | Rater 2 | Score |
|---|---|---|---|---|
| 62 | - An image is positive if it is a continuous, single-line drawing with smooth, flowing lines and no sharp angles or uncontrolled intersections/self-crossings. An image is negative if it includes sharp angles, abrupt changes in direction, disconnected segments, or uncontrolled intersections/self-crossings. | 0 | 0 | 0 |
| 63 | An image is positive if it contains two boundaries where the inner boundary is fully enclosed by the outer boundary, and the inner and outer boundaries are similar in shape. Otherwise, the image is negative. | 0 | 0 | 0 |
| 64 | - An image is **positive** if the small circle and the "+" or "x" symbol are on **opposite sides** of the ellipse. - An image is **negative** if the small circle and the "+" symbol are on the **same side** of the ellipse. | 0 | 0 | 0 |
| 65 | An image is positive if and only if the number of triangles equals the number of circles. Otherwise, the image is negative. | 0 | 0 | 0 |
| 66 | An image is positive if the unconnected circles are not aligned in a straight vertical line. An image is negative if the unconnected circles are aligned in a straight vertical line. | 1 | 1 | 1 |
| 67 | Positive images typically have two curved lines that intersect exactly once. The intersection is clean, and the lines do not touch or overlap at any other point. Negative images typically have two curved lines that either do not intersect at all (e.g., they are disjoint or only touch at the endpoints) or intersect more than once. An image is positive if the two curved lines intersect exactly once. Otherwise, the image is negative. | 0 | 0 | 0 |
| 68 | An image is positive if the lines intersect at a single point and do not cross each other again. An image is negative if any of the curves cross another line after the initial intersection. | 0 | 0 | 0 |

| Problem Index | Rule | Rater 1 | Rater 2 | Score |
|---|---|---|---|---|
| 69 | - An image is positive if at least one branch crosses the vertical line. Otherwise, the image is negative. | 0 | 0 | 0 |
| 70 | An image is positive if at least one pair of lines crosses each other. An image is negative if no lines cross each other. | 0 | 0 | 0 |
| 71 | An image is positive if and only if it contains at least one shape nested inside another shape, and the nested shapes are of different types. Otherwise, the image is negative. | 0 | 0 | 0 |
| 72 | A shape is positive if it is a single continuous line that is smooth, flowing, open (not a closed loop), and does not intersect itself. A shape is negative if it has sharp angles, is a closed loop, or has self-intersecting lines. | 0 | 0 | 0 |
| 73 | - An image is positive if the shapes are not aligned in a straight line. - An image is negative if the shapes are aligned in a straight line or have a structured arrangement. | 0 | 0 | 0 |
| 74 | An image is positive if it contains a sharp, pointed end. An image is negative if it lacks a sharp point and is predominantly rounded or looped. | 0 | 0 | 0 |
| 75 | An image is positive if the curved line points toward the triangle or interacts with it. An image is negative if the curved line does not point toward the triangle or does not interact with it. | 1 | 0 | 0 |
| 76 | An image is positive if it contains at least one concave corner and is asymmetric. An image is negative if it has no concave corners, is symmetric, or forms a closed loop. | 0 | 0 | 0 |
| 77 | An image is positive if all the lines originate from a single point, forming a "fan" or "V" shape. An image is negative if the lines do not all originate from a single point. | 0 | 0 | 0 |
| 78 | - An image is positive if no two lines are parallel. - An image is negative if at least two lines are parallel. | 0 | 0 | 0 |

*Continued on next page*

| Problem Index | Rule | Rater 1 | Rater 2 | Score |
|---|---|---|---|---|
| 79 | - An image is positive if the black circle and white circle are adjacent. Otherwise, the image is negative. | 0 | 0 | 0 |
| 80 | An image is positive if the "+" symbol is positioned between the two black dots, near the midpoint of the line segment connecting them. Otherwise, the image is negative. | 0 | 0 | 0 |
| 81 | An image is positive if it contains at least one filled shape and one unfilled shape. An image is negative if it contains only filled shapes or only unfilled shapes. | 0 | 0 | 0 |
| 82 | An image is positive if the circle does not overlap with any of the plus signs. An image is negative if the circle overlaps with at least one plus sign. | 0 | 0 | 0 |
| 83 | An image is positive if it contains exactly one circle, surrounded by exactly four plus signs arranged symmetrically around the circle. Otherwise, the image is negative. | 0 | 0 | 0 |
| 84 | - An image is positive if the square is **outside** the arrangement of circles. - An image is negative if the square is **inside** the arrangement of circles. | 1 | 1 | 1 |
| 85 | - An image is positive if it does not contain any closed shapes or regions. - An image is negative if it contains at least one closed shape or region. | 0 | 0 | 0 |
| 86 | An image is positive if it contains exactly one branching point where exactly three lines meet. An image is negative if it contains a branching point where four or more lines meet or if it has additional intersections or overlapping lines. | 1 | 1 | 1 |

| Problem Index | Rule | Rater 1 | Rater 2 | Score |
|---|---|---|---|---|
| 87 | An image is positive if it contains either: 1. A single connected structure formed by lines, or 2. A closed shape with exactly four sides (quadrilateral). An image is negative if it contains: 1. Multiple disconnected components, or 2. A closed shape with more than four sides, or 3. Symmetrical intersecting lines (e.g., "plus" or "cross"). | 0 | 0 | 0 |
| 88 | An image is positive if all the shapes are connected into a single structure. An image is negative if the shapes are separate or overlapping without merging. | 0 | 0 | 0 |
| 89 | An image is positive if and only if every black-filled shape has a corresponding hollow shape, and they are vertically aligned in a 1-to-1 correspondence. Otherwise, the image is negative. | 0 | 0 | 0 |
| 90 | An image is positive if all groups of shapes alternate between white and black and have an even number of shapes. An image is negative if any group of shapes does not alternate consistently or has an odd number of shapes. | 0 | 0 | 0 |
| 91 | An image is positive if it contains asymmetry or irregularity in its structure. An image is negative if it exhibits symmetry, regularity, or uniformity in its structure. | 0 | 0 | 0 |
| 92 | An image is positive if it contains exactly one filled dot that is smoothly integrated into a continuous shape without disrupting its flow or creating intersections. Otherwise, the image is negative. | 0 | 0 | 0 |
| 93 | An image is positive if it contains an asymmetric, wavy, or oscillatory pattern without forming closed shapes. An image is negative if it contains symmetry, closed shapes, or static geometric arrangements. | 0 | 0 | 0 |
| 94 | An image is positive if the black circle is centrally located within the arrangement of circles. An image is negative if the black circle is at the edge of the arrangement. | 1 | 1 | 1 |

| Problem Index | Rule | Rater 1 | Rater 2 | Score |
|---|---|---|---|---|
| 95 | An image is positive if the lines inside the shape are vertical. An image is negative if the lines inside the shape are horizontal. | 1 | 1 | 1 |
| 96 | An image is **positive** if the lines converge or taper to form a shape with a clear focal point. An image is **negative** if the lines are parallel, grid-like, or do not converge to a single point. | 0 | 0 | 0 |
| 97 | - An image is positive if it contains a triangular shape. - An image is negative if it contains a circular shape. | 1 | 1 | 1 |
| 98 | An image is positive if it contains at least one triangle. Otherwise, it is negative. | 1 | 1 | 1 |
| 99 | An image is positive if the green circles and black triangles are intermixed. An image is negative if the green circles and black triangles are segregated into separate regions. | 0 | 1 | 0 |
| 100 | An image is positive if it resembles the letter "A" (angular, triangular central region, horizontal bar). An image is negative if it resembles the letter "B" (rounded, two loops, vertical line). | 1 | 1 | 1 |
| **Total** | | **30** | **34** | **25** |

## B PROMPTS

### B.1 PROGRAM SYNTHESIS PROMPTS

In order to ensure that the programs synthesized by the VLM are able to manipulate the correct objects, we prompt the VLM to identify some important objects and transform those into method stubs. These suggested methods are intended to provide guidance on what final programs should look like (i.e., what parts of the image should be focused on / manipulated by the programs)

```
----- user -------------------
You are solving a Bongard-style problem where you must write a program that outputs
 'POSITIVE' if an input image is an example of the positive concept <positive
concept> and 'NEGATIVE' otherwise.
Consider the steps you must take to write this program.
List 0-3 objects you will need to detect in the image. Please output as a comma-
separated list in the format <objects>object1, object2, object3</objects>.
```

```
Example: The positive concept is 'many squares' and the negative concept is 'few
squares'.

Answer:
<objects>square</objects>
'''
```

The returned list of objects are transformed into method stubs with the following signature:

```python
def find_<object_name>(image: np.ndarray) -> np.ndarray:
    """
    Returns the bounding boxes for all {obj}s in the image, if they exist, and None
 otherwise. The output array has the shape [N, 4] where N is the number of {obj}s
in that image, and 4 corresponds to the bounding box format [x coordinate of upper
left hand corner,
    y coordinate of upper left hand corner, width of the box, height of the box].
    """
```

These are used in program generation prompt, which uses the template below:

```
----- user --------------------
You are solving a Bongard-style problem where you will be given several examples of
 two hidden concepts, along with the rule for each of these examples. Your job is
to write a Python program that will determine whether an input image is a positive
or negative example of a concept. This program must generalize to images other than
 the examples I give you. These are the positive examples, which represent the
concept <positive concept>.

<positive examples>

----- system --------------------
I see you've uploaded the positive examples. Please upload the negative examples.

----- user --------------------
These are the negative examples, which represent the concept <negative concept>
Please structure your program as a detection phase, where you first detect the
necessary objects in the image, and then a classification phase, where you perform
a series of operations to determine whether each image is a positive or negative
example. The following method stubs are given to you as suggestions of methods you
might want to implement for the detection phase: <existing method stubs>

<negative examples>

----- system --------------------
Please provide instructions for the program I need to write.

----- user --------------------
Please write <n_programs> different Python programs, each enclosed in Markdown
backticks, that will determine whether an input image is an example of the positive
```

```
 concept {positive_concept}. Each program should include a method called
classify_image that, given an input image test_image, as well as any parameters
that you needed to use in your program, will correctly output a 'POSITIVE' or '
NEGATIVE' classification. If helper methods other than those given to you are
needed, please fully implement them all.  Please also make a comment specifying
 the range of values for any parameters to the function, in the format 'values(
param_name):(low, high)', and a comment specifying the type of the parameter as
either an int or a float in the format 'type(param_name): int' or 'type(param_name)
: float', respectively.  Think a bit before you start writing code.

----- system -------------------
Please provide an example of how to generate these programs.

----- user -------------------
<retrieved examples>
```

Programs that do not achieve a perfect score on the training set after optimization are repaired using this prompt. Note that the message about an exception is only displayed if an exception occurred.

```
----- user -------------------
You are an expert Python programmer. You wrote the following program: <program>"""

When running the program, the following exception was encountered: <exception>

The program returned the wrong output on <# positive examples classified
incorrectly> images that were positive examples of the concept <positive concept>
and <# negative examples classified incorrectly> images that were negative examples
.

Please output a repaired version of this program enclosed in Markdown backticks.
You are able to use libraries like OpenCV, numpy, and scipy.

Please also make a comment specifying the range of values for any parameters to the
 function, in the format 'values(param_name):(low, high)'
, and a comment specifying the type of the parameter as either an int or a float in
 the format 'type(param_name): int' or 'type(param_name): float', respectively.
Think a bit about what went wrong with the original implementation before you start
 writing code.
```

## B.2 PROMPTS FOR VLM VERIFICATION AND SOLUTIONS

The following prompt template was used for the verification task with the base VLMs, as well as when the VLMs were called inside our program+CoT verifier:

```
----- user -------------------
You are solving a Bongard-style problem where you need to check whether an image
corresponds to the rule <positive concept>, which separates positive and negative
images. The negative images adhere to the rule <negative concept> instead.
```

```
Here are <n_shot> positive examples. Please look at them and then await the
negative examples, which I will give you after this message. Answer with only ok
and nothing else.
-----  system  -------------------
      ok.
-----  user  -------------------
Here are <n_shot> negative examples. These do not fulfill the rule <positive
concept>, but instead adhere to the rule <negative concept>. Please look at them,
and then, finally, I will give you a last image which you should classify as
positive (adheres to the positive rule) or negative (does not adhere to the
positive rule, but instead to the negative rule). Answer with only ok and nothing
else.
-----  system  -------------------
      ok.
-----  user  -------------------
Taking all prior information into consideration, given the following image, do you
think it is positive, meaning it displays the concept '<positive concept>'? Or is
it negative and displays the concept '<negative concept>'? First think about it,
and then provide your answer in the following form:
Output enclosed in Markdown backticks either POSITIVE or NEGATIVE depending on your
 final decision. Do not produce any other output.
```

Hypotheses were generated using the prompt below:

```
-----  user  -------------------
You are solving a Bongard-style problem where you will be given several examples of
 two hidden concepts, along with the rule for each of these examples. Your job is
to write a Python program that will determine whether an input image is a positive
or negative example of a concept. This program must generalize to images other than
 the examples I give you. These are the positive examples, which represent the
concept <positive concept>.

<positive examples>

-----  system  -------------------
I see you've uploaded the positive examples. Please upload the negative examples.

-----  user  -------------------
These are the negative examples, which represent the concept <negative concept>
Please structure your program as a detection phase, where you first detect the
necessary objects in the image, and then a classification phase, where you perform
a series of operations to determine whether each image is a positive or negative
example. The following method stubs are given to you as suggestions of methods you
might want to implement for the detection phase: <existing method stubs>

<negative examples>

-----  system  -------------------
I see you've uploaded the negative examples. Please provide instructions for
solving the Bongard problem.
```

```
----- user --------------------
Given these positive and negative images, please do the following:
1. Someone has given you the following rules: <example rules>. Consider how these
rules apply to the positive and negative examples. Which examples do each of them
work on? Which examples do they fail on?
2. Output <n_sample> rules which predict when an image is positive. Please enclose
each rule in <rule></rule>, e.g. <rule>contains red circle</rule>"""
```

Baseline VLMs were tested on the solution task using the following prompt:

```
----- user --------------------
You are solving a Bongard-style problem where to solve the problem you need to
infer a hidden rule that separates positive and negative images. Pay attention to
abstract geometric properties.
Here are <n_shot> positive examples. Please look at them and then await the
negative examples, which I will give you after this message.

<positives>

----- system --------------------
I see you've uploaded the positive examples. Please provide the negative examples
for the Bongard problem, and I'll help you analyze the differences between the two
groups in order to infer the hidden rule that separates positive and negative
images.

----- user --------------------
Here are <n_shot> negative examples.

1. Analyze the positive examples (looking for what is common between them)
2. Analyze the negative examples (looking for what is common between them)
3. Compare negative and positive examples (looking for what is different between
them)
4. Output a rule which predicts when an image is positive or negative.

<negatives>
```

## C    VERIFIER DETAILS FOR SOLUTION TASK

When using our verifier in the solution task, we make two slight modifications to our original system. These modifications were made because the number of rules to verify in the solution task is much larger than the verification task, where there was only one rule per problem.

1. Programs generated per rule is set to 5 instead of 10

2. Rather than generating programs and evaluating on 6 different train/test splits, we only evaluate on one, and the evaluation score along with the training score is the accuracy assigned to the rule

## D  RAG AND GENERATED PROGRAMS

We generated a dataset of 59 BP programs by either writing programs by hand or editing the output of older versions of our verifier. Sample programs for BP # 14 ('large total line length' vs. 'small total line length') and BP #40 ('three points collinear') are included below. Since the BP problems are hand-drawn, problems like BP#40 are actually a matter of finding *approximately* collinear points.

```python
import cv2
import numpy as np
from typing import List

def find_lines(image):
    """
    Detects lines in the image and returns their contours
    """
    if len(image.shape) > 2 and image.shape[2] > 1:
        gray = cv2.cvtColor(image, cv2.COLOR_BGR2GRAY)
    else:
        gray = image.copy()

    # Threshold the image to get binary image
    _, binary = cv2.threshold(gray, 127, 255, cv2.THRESH_BINARY_INV)

    # Find contours in the binary image
    contours, _ = cv2.findContours(binary, cv2.RETR_EXTERNAL, cv2.
CHAIN_APPROX_SIMPLE)

    return contours

def calculate_line_length(contour):
    """
    Calculate the approximate length of a line represented by a contour
    """
    # For a line, the perimeter is approximately twice its length
    perimeter = cv2.arcLength(contour, closed=False)
    return perimeter / 2

def classify_image(image, length_threshold=500):
    """
    Classifies an image as 'POSITIVE' if it has large total line length, 'NEGATIVE'
 otherwise.

    Args:
        image: the image to classify
        length_threshold: Threshold for total line length to be considered "large"
        values(length_threshold): (100, 2000)
        type(length_threshold): float

    Returns:
        A 'POSITIVE' or 'NEGATIVE' classification for the image
    """
    contours = find_lines(image)
```

```
        # Calculate total line length in the image
        total_length = 0
        for contour in contours:
            length = calculate_line_length(contour)
            total_length += length

        # Normalize by image size
        image_diagonal = np.sqrt(image.shape[0]**2 + image.shape[1]**2)
        normalized_length = total_length / image_diagonal

        # Classify based on normalized length
        if normalized_length > length_threshold / 1000:  # Convert to reasonable scale
            return 'POSITIVE'
        else:
            return 'NEGATIVE'
```

```
import numpy as np
import cv2
from scipy import ndimage

def find_points(image):
    """
    Find all points in the image by detecting contours.
    Returns a list of (x, y) coordinates representing the centers of objects.
    """
    if (len(image.shape) == 3):
        gray = cv2.cvtColor(image, cv2.COLOR_BGR2GRAY)
    else:
        gray = image.copy()
    (_, binary) = cv2.threshold(gray, 127, 255, cv2.THRESH_BINARY_INV)
    (contours, _) = cv2.findContours(binary, cv2.RETR_EXTERNAL, cv2.
CHAIN_APPROX_SIMPLE)
    points = []
    for contour in contours:
        M = cv2.moments(contour)
        if (M['m00'] != 0):
            cx = int((M['m10'] / M['m00']))
            cy = int((M['m01'] / M['m00']))
            points.append((cx, cy))
    return points

def has_three_collinear_points(points, slope_tolerance, distance_threshold):
    """
    Check if there are at least 3 collinear points.

    Args:
    points: List of (x, y) coordinates
    slope_tolerance: Maximum allowed difference in slopes to consider lines
    parallel
```

```
        distance_threshold: Maximum distance from a point to a line to consider it
    collinear

        Returns:
        True if at least 3 points are collinear, False otherwise\n
        """
        n = len(points)
        if (n < 3):
            return False
        for i in range(n):
            for j in range((i + 1), n):
                (x1, y1) = points[i]
                (x2, y2) = points[j]
                collinear_points = [points[i], points[j]]
                if (abs((x2 - x1)) < 1e-06):
                    (a, b, c) = (1, 0, (- x1))
                else:
                    slope = ((y2 - y1) / (x2 - x1))
                    a = slope
                    b = (- 1)
                    c = (y1 - (slope * x1))
                norm = np.sqrt(((a * a) + (b * b)))
                (a, b, c) = ((a / norm), (b / norm), (c / norm))
                for k in range(n):
                    if ((k != i) and (k != j)):
                        (x3, y3) = points[k]
                        distance = abs((((a * x3) + (b * y3)) + c))
                        if (distance < distance_threshold):
                            collinear_points.append(points[k])
                            if (len(collinear_points) >= 3):
                                return True
        return False

    def classify_image(image, slope_tolerance=0.05, distance_threshold=2.0):
        """
        Classify an image based on whether they contain at least 3 collinear points.

        Args:
        image: image to classify
        slope_tolerance: Tolerance for slope differences
        distance_threshold: Maximum distance to consider a point collinear

        Returns:
        'POSITIVE' or 'NEGATIVE' classification
        """
        # values(slope_tolerance): (0.01, 0.1)
        # type(slope_tolerance): float
        # values(distance_threshold): (1.0, 5.0)
        # type(distance_threshold): float
        points = find_points(image)
        if ((points is not None) and (len(points) >= 3)):
            if has_three_collinear_points(points, slope_tolerance, distance_threshold):
                return 'POSITIVE'
```

```
        return 'NEGATIVE'
```

# E   ADDITIONAL TECHNICAL DETAILS

We use the high resolution BP dataset introduced in Depeweg et al. (2024).

For all verification experiments, the number of programs sampled per rule is 10. This is decreased to 5 for solution experiments. The number of rules sampled per problem is always 6.

For optimization, we perform 15 iterations of Bayesian optimization.

We sample natural language rules at temperature 1 and code at temperature 0.5.

