# OpenReview forum: "Bongards at the Boundary of Perception and Reasoning: Programs or Language?"
_ICLR.cc/2026/Conference — ICLR 2026 Conference Withdrawn Submission_

### Official Review · Reviewer_Yk3o · 2025-10-16

**Soundness:** 1
**Presentation:** 1
**Contribution:** 1
**Rating:** 2
**Confidence:** 4

**Summary:**

These paper aims to investigate in how far natural language statements from VLMs help to solve Bongard Problems in contrast to program-based rules. The authors propose a specific setup to combine both forms of rules and to evaluate these.

**Strengths:**

I think the general research question is valid and interesting: are natural language statements or programs or both more valuable to solving such visual puzzles via VLMs.

**Weaknesses:**

While I find value in the research question, comparing natural language versus program representations for solving business processes (BPs), and appreciate its conceptual simplicity, the paper suffers from significant gaps in critical information. Specifically, the authors fail to clearly articulate both their contribution (which appears to be a prompting pipeline, if I understand correctly) and its mechanics. Equally problematic is the lack of detail regarding their experimental methodology. The numerous questions and concerns I encountered while reading (detailed in the questions section) make it extremely difficult to validate the conclusions the authors have drawn.

I think the paper could also improve from making the citation style more consistent, e.g., publication venues are missing for some of the references.

**Questions:**

I will just post these questions in chronological order:

ll 137 and on: The reader needs more context upfront in order to understand what the role is of these two tasks. Importantly, how are these two tasks depicted in the results?

Figure 2 seems to be missing the core parts that are also missing in the text (see questions below): in 2B) what is happening under "VLM and/or python" and "scoring on example images"? How are these things being done? I think this is the most important part of the figure, but neither the figure nor the text explains this.

How are the natural language hypotheses ever used in the further pipeline that I believe are introduced in section 4.1.1?

(Related to the above) ll 181: "which combines natural language and Python programs" This is very ill-specified. What does the verifier do exactly? What is the verifier? Where do the programs come from? How does it integrate the language rules?

Figure 3: How are programs "sampled"?

ll 214: What form do the programs have? Are they python programs? How are the programs evaluated on the images?

Section 5: What is the experimental setup? "all three tasks" (ll 279) What three tasks? How are the models evaluated? Who are these human raters? What was the setup for their judgement? What images are the authors testing on? It seems all are being used to find the programs and at the same time evaluate the performance of these?

Table 1: What does the Verification value tell us? Is this the percent of solved tasks overall or the percent of correctly classified test images per BP? Why are values for "+ programs" missing? Again, how does the "both" setup work?

Section 5.2: Where exectly do we see the CoT performance? Is this just the base model's performance?

Section 5.3.1: Where do these human numbers come from? What study was this? How was it performed?

Figure 5: Why does the figure now say LLM instead of VLM? What LLM/VLM was used in these specific comparisons?

ll 373: Is the goal to model human problem solving skills? Couldn't they be solving these tasks completely differently?

Section 5.3.2: What is the intuition behind whether this simple form of inversion is sufficient to test for memorization?

Minor: I'm not sure what the benefit is of showing Raven examples in Figure 1.

---

### Official Review · Reviewer_VZiA · 2025-10-22

**Soundness:** 1
**Presentation:** 1
**Contribution:** 3
**Rating:** 2
**Confidence:** 4

**Summary:**

The paper presents a neurosymbolic approach for solving Bongard problems using VLMs. It combines rule-based reasoning and program synthesis, where hypothesized visual rules are represented as parameterized programs and optimized using Bayesian methods. The work aims to bridge symbolic reasoning and neural perception in visual problem-solving.

**Strengths:**

- The topic is interesting and timely, addressing the gap in reasoning in current VLMs.
- Using programmatic representations to reason about visual rules is a creative idea that connects symbolic reasoning with modern VLMs.

**Weaknesses:**

I provide a high-level list of my concerns, and detailed questions for clarification and suggestions are listed in the Question section:
* **Methodology lacks clarity** — key processes like verification, rule generation, and program execution are not clearly described or connected.
* **Experimental design is confusing** — sampling choices, baselines, and result interpretation are not well justified.
* **Presentation is hard to follow** — the paper needs a clearer structure, figures, and explanations.
* **Limited experiments** — The evaluation of the proposed methodology is very limited and only focuses on Bongard problems, leaving the question of whether it generalizes outside the Bongard domain.

**Questions:**

1. Hypthesis generation (Section 4.1.1)
-The rationale behind sampling 6 rules per problem (and another 3 random rules) is missing.
- As a baseline comparison, it would be interesting to compare your methodology to some basic prompting of the VLM to select one of the rules. E.g., the rule it considers to be the best fit for the Bongard problem. This way, you have similar computational efforts and can show real improvements in using your Bayesian parameter optimization technique.

2. Verification Process – Unclear (Section 4.1.2)
-Section 4.1.2 (Verifier) is difficult to follow. It’s not clear how images are processed, how candidate programs are verified, or how images are assigned labels from the candidate rule.
-The relation between 4.1.2 Verifier and 4.1.3 Verification with Programs is ambiguous. It’s unclear whether 4.1.3 is part of the verifier or an independent stage.

3. Program and Embedding Similarity (4.1.3)
-The paper uses embedding similarity (line 226) between generated and ground-truth rules to select examples, but the embedding method is unspecified.
-Logical similarity is not well represented by embeddings—small changes (e.g., adding a not) can completely invert rule behavior.

4. System Integration (Section 4.1 overall)
-Sections 4.1.1–4.1.4 are presented as separate modules, but their integration is not described.
-A diagram summarizing how these modules connect would help readers understand the full pipeline.

5. Memorization Experiment (Section 5.3.2)
-The “Inversion” explanation (Section 5.3.2) appears too late—it should be introduced when the table first appears.
-The use of label inversion as a test for memorization is questionable (section 5.3.2). Simply swapping class labels does not necessarily distinguish memorization from reasoning.

6. Concept Acquisition Discussion (Section 5.5)
-The final claim that AI systems can “acquire and use new concepts” (line 424) is not well supported.
-The authors should better explain how this differs from traditional VLM behavior.

7. Additional Ambiguities in Setup
-The choice to synthesize 10 programs per problem (line 310) is unexplained. What exactly do you mean by that?
-Figure 3 is never referenced or discussed.
-Table 1 lacks clarity: what is the difference between model, model+programs, and model+both?
-The CoT baseline (line 313) mentioned in the text is not labeled in Table 2.
-The statement “While this is informative, it leaves open the question of whether the stated rule actually governs the model’s predictions” (lines 410–411, section 5.4) should be clarified. What do you mean by the stated rule and the model's prediction?

8. Missing Context and Comparisons
- An additional comparison to a reasoning model would be interesting. In this regard you might want to compare to Wüst et al.’s Bongard in Wonderland study "https://ml-research.github.io/bongard-in-wonderland/" . Reasoning models (o3, GPT-5) solve 53 and 64 Bongard tasks, respectively.
-Including this would contextualize performance. While they do outperform your methodology, it is fair to assume that model sizes and computational costs are much larger. A computational performance tradeoff might be interesting and strengthen your results.
- An additional evaluation on other reasoning problems outside the Bongard domain would strengthen the empirical results of the paper.

9. Writing and Presentation
-The writing could be simplified and clarified, especially in Sections 4 and 5, see question and suggestions for improvement above.
-Clearer subheadings, schematic figures, and consistent terminology would improve readability.

---

### Official Review · Reviewer_cE2d · 2025-11-06

**Soundness:** 1
**Presentation:** 2
**Contribution:** 3
**Rating:** 2
**Confidence:** 4

**Summary:**

The paper presents a neuro-symbolic framework for solving Bongard problems that combines a vision-language model (VLM) with a verifier. The VLM generates natural language hypotheses about the underlying rule of a given Bongard problem. These hypotheses are then passed to the verifier, which uses both natural language and executable Python programs to test whether the hypotheses hold on the training examples. The verifier adjusts the parameters of the programs using Bayesian optimization to maximize their fit to the training set. The proposed approach shows improvements over baseline model-only methods.

**Strengths:**

- Combining a VLM’s ability to propose hypotheses with a partly symbolic verifier is a compelling idea that seems like a good direction towards solving Bongard problems
- The use of Bayesian optimization to tune the parameters of the generated programs is a technically sound and interesting contribution.

**Weaknesses:**

- The exposition is often confusing and lacks precision in several key parts. For example, lines 166–170 make it unclear when the verifier receives ground truth information and what exactly constitutes the solution task. Similarly, lines 226–229 do not clearly explain whether one or multiple programs are used, and how these are sampled.

- The paper’s terminology for different task components is confusing, making it difficult to follow the workflow. For example, the method section begins by introducing two tasks, but in the experiments, three tasks are mentioned (line 279) (supposedly the inversion task?), which was not previously mentioned. Also, the use of verifier vs. verifier task is a bit confusing, since the verifier is relevant for both tasks.

- The input to the verifier could be elaborated on in more detail, e.g., the method stubs are only mentioned once, and it remains unclear how they relate to the implementation of the programs.

- The approach appears to rely heavily on prior knowledge, including pre-existing solutions to Bongard problems and prior program structures. This reduces the perceived level of generalization and makes comparisons to human reasoning less fair.  It would be important to discuss the extent to which it introduces excessive background knowledge into the problem setup (and evaluate how the method performs when it is not provided). While a fixed grammar for the verifier across all problems would be a reasonable design choice, using distinct solution programs for each case risks providing too much task-specific prior knowledge and undermining the generality of the approach.

Overall, the paper proposes an interesting neuro-symbolic method that integrates VLM-generated hypotheses with program verification and Bayesian optimization. While the conceptual contribution is potentially impactful, the presentation is confusing, the scope of experiments is limited, and the reliance on prior knowledge raises questions about fairness and generalization. Clarifying the workflow, justifying the design decisions, and expanding the experimental evaluation would significantly strengthen the paper.

**Questions:**

- Q1) In Table 2, programs appear to perform less successfully than expected. Could the authors provide intuition for why this occurs?

- Q2) Could the authors quantify and categorize the failure cases? Specifically, is the main source of error that the VLM fails to generate correct hypotheses, or that the symbolic program/language cannot represent them accurately?

- Q3) A few concrete examples of hypotheses, resulting programs, and verification outcomes would make the approach clearer and more interpretable.

- Q4) Have the authors explored using other vision-language models (e.g., InternVL3 or Qwen2.5-VL) to test whether smaller or differently aligned models yield similar improvements?

---

### Official Review · Reviewer_Y2uZ · 2025-11-06

**Soundness:** 1
**Presentation:** 2
**Contribution:** 1
**Rating:** 2
**Confidence:** 4

**Summary:**

This paper presents a framework designed to evaluate VLM capabilities on Bongard Problems (BP). The proposed system has two main components: one that uses VLM to generate several possible rules in natural language, and another that verifies the proposed rules. The verifier has two versions. One uses VLM for Python program synthesis, a parameter sampling and Bayesian optimization module, and a final VLM judgment. The second version also uses VLM and some labeled examples for few-shot CoT reasoning to determine the correct rule if program synthesis fails. The authors tested three approaches with two VLMs (GPT-4o and Claude 3.7 Sonnet) on two tasks: BP solving and BP verification. They reported moderate improvement with CoT on the BP solving task, though performance worsened with program synthesis and CoT as a backup. There were slight overall improvements on the BP verification task, but inconsistent performance in different BP categories.

**Strengths:**

This paper presents an interesting approach to solving Bongard problems using program synthesis by VLM, in combination with function optimization methods.

**Weaknesses:**

Unfortunately, this paper has several weaknesses regarding its contribution, the soundness of the proposed method, its clarity, and the discussion of limitations:

1.  It is unclear whether the proposed method will be considered useful. The neurosymbolic method uses parameterized programs and, according to the abstract, is a way to overcome the limitations of current VLMs when solving Bongard problems. However, the evaluations do not support or validate this expectation. The paper is based on the expectation that program synthesis will meaningfully improve performance or at least provide relevant insights into the limitations of current VLMs. Without CoT backup (VLM + programs), however, the performance of the program synthesis approach is worse for BP verification, and it is not reported for BP solutions.
2.  At least the very important ablation study, which would have used few-shot CoT on the VLM with suggested rules and no formal program synthesis, was not performed. Consequently, the reported results and performance improvements are difficult to interpret, offering limited insight into the effectiveness of the function optimization approach.
3.  Furthermore, it is not discussed or evaluated to what extent the improvement in performance is reliant on the additional information that is provided, A) during the few-shot rule candidate generation when providing 12 ground truth rules of previous BP, and B) within the verification process using embedding similarity to the selection of 59 ground truth programs. It should also be noted that this makes it difficult to compare with the human results.
4.  The paper lacks a discussion of possible and evidential limitations of their method and their evaluation scheme. It should be clear at which steps of the rule generation or verification process the method relies on or tests different VLM capabilities, such as visual description, reasoning, code generation, and visual question answering. These capabilities have different limitations that may impact the performance and interpretation of the proposed evaluations, which should be addressed within the setup/methods section and/or a proper discussion or limitations section.

**Questions:**

Addressing the following questions and comments could improve the clarity of the paper, in addition to resolving the issues raised above:

1.  It is slightly confusing that on p.3, there are two tasks defined when Table 1 and Section 5.1 show three tasks.
2.  Following the exact execution of the method(s) is quite difficult, particularly with regard to integrating external information with "training examples" and selecting the best candidate programs using embedding similarity to some ground truth.
    - Some related wording: The 6 rules of the three previous BP are not "sampled", as you take all 6 (two per BP)
    - How were the "59 problems" (4.1.3) chosen? And why were they selected? Does this not completely invalidate any conclusion in terms of the proposed method's BP-solving capabilities? It seems that, if the introduced behavioral bias is not properly discussed, the method becomes purely diagnostic.
3.  Why is there no evaluation of VLM+programs on the BP solving task?
4.  Because the proposed evaluation framework combines methods that rely on different types of external information or VLM capabilities, a separate ablation study for each method would be a good practice. This would determine the effectiveness of each method in the proposed setup and identify its limitations.

---

### Note · Authors · 2025-11-19

I have read and agree with the venue's withdrawal policy on behalf of myself and my co-authors.